# Aging aggravates aortic aneurysm and dissection via miR-1204-MYLK signaling axis in mice

Ze-Long Liu [1,2,3,4,5,15], Yan Li[1,2,3,4,5,15], Yi-Jun Lin[1,2,3,4,5,15], Mao-Mao Shi[1,2,3,4,5,15], Meng-Xia Fu[1,2,3,4,5,15], Zhi-Qing Li[6,7], Da-Sheng Ning[1,2,3,4,5], Xiang-Ming Zeng[1,2,3,4,5], Xiang Liu[1,2,3,4,5], Qing-Hua Cui[8], Yue-Ming Peng[1,2,3,4,5], Xin-Min Zhou[9], Ye-Rong Hu[9], Jia-Sheng Liu[1,2,3,4,5], Yu-Jia Liu [1,2,3,4,5], Mian Wang[2,10], Chun-Xiang Zhang [11,12], Wei Kong [6,7] ✉, Zhi-Jun Ou [2,3,4,5,13] ✉ & Jing-Song Ou [1,2,3,4,5,14] ✉

The mechanism by which aging induces aortic aneurysm and dissection (AAD) remains unclear. A total of 430 participants were recruited for the screening of differentially expressed plasma microRNAs (miRNAs). We found that miR-1204 is significantly increased in both the plasma and aorta of elder patients with AAD and is positively correlated with age. Cell senescence induces the expression of miR-1204 through p53 interaction with plasmacytoma variant translocation 1, and miR-1204 induces vascular smooth muscle cell (VSMC) senescence to form a positive feedback loop. Furthermore, miR-1204 aggravates angiotensin II-induced AAD formation, and inhibition of miR-1204 attenuates β-aminopropionitrile monofumarate-induced AAD development in mice. Mechanistically, miR-1204 directly targets myosin light chain kinase (MYLK), leading to the acquisition of a senescence-associated secretory phenotype (SASP) by VSMCs and loss of their contractile phenotype. MYLK overexpression reverses miR-1204-induced VSMC senescence, SASP and contractile phenotypic changes, and the decrease of transforming growth factor-β signaling pathway. Our findings suggest that aging aggravates AAD via the miR-1204-MYLK signaling axis.

Aortic dissection (AD) and aneurysms, including thoracic aortic aneurysms (TAAs) and abdominal aortic aneurysms (AAAs), are potentially fatal diseases[1,2]. In particular, ascending AD is associated with a mortality rate of 50% within two days and 72% within one week. The 1-year survival rate is dismal 10% without surgical intervention. Moreover, no preventive or predictive methods exist for aortic aneurysm and dissection (AAD). The incidence of AAD has been shown to be higher in older adults, suggesting that aging is a risk factor for AAD[3–5]. However, the reason for the higher incidence of AAD in older adults than in younger adults remains unclear.

Recent studies have shown that aging may affect AAA formation[6–9]. For example, the expression of sirtuin 1 is decreased in both human and mouse AAA samples, and the reduction in sirtuin 1 increases vascular smooth muscle cell (VSMC) senescence and AAA formation[6,7]. Another study showed that the expression of myocardin-related transcription factor A is increased in both human and mouse thoracic and abdominal aneurysmal tissues and that the deletion of myocardin-related transcription factor A reduces VSMC senescence and AAA formation[8]. Similarly, a lack of Kruppel-like factor 5 induces vascular senescence, which correlates with AAA rupture[9]. VSMC senescence has also been

observed in mouse and human thoracic aortic aneurysm and dissection (TAAD) specimens[10]. Although these studies have demonstrated a relationship between VSMC senescence and AAD, the mechanism by which aging induces AAD remains unknown.

Previous studies have demonstrated that micro-ribonucleic acids (miRNAs) regulate cell senescence and inflammation, and play a critical role in aging-related diseases[11]. Additionally, aging induces miRNA expression[12,13]. MiRNAs are also involved in AAD formation[14,15]. However, it remains unknown whether miRNAs induce vascular senescence, thereby promoting AAD formation through the induction of inflammation. In the present study, we found that miR-1204 was markedly upregulated in the plasma and aorta of older patients with AAD, including those with TAA and AAA. Aging induces miR-1204, which inhibits myosin light chain kinase (MYLK), facilitating the acquisition of the senescence-associated secretory phenotype (SASP) and the loss of the contractile phenotype in VSMCs. This results in cytokine/chemokine release, vascular inflammation, and VSMC dedifferentiation, which aggravates AAD formation. Furthermore, miR-1204 is induced through the interaction of p53 with the plasmacytoma variant translocation 1(PVT1) response element in senescent VSMCs. miR-1204 induces aging by forming a positive feedback loop that aggravates AAD formation.

## Results

### miR-1204 is upregulated with aging in patients with AAD

To identify aging-associated miRNAs involved in AAD, 430 participants were recruited for plasma miRNA detection. The participants were divided into four groups based on age and disease state: namely young normal (<50 years old), elder normal (>50 years old), young patient (<50 years old), and elder patient (>50 years old) based on their age and disease states. We selected 50 years as the cutoff for the groups because the average age of patients with AD in China is less than 50 years, which is significantly different from that in Western countries (> 60 years)[16]. The demographic data and basic clinical parameters are presented in Table 1 and Supplemental Table 1. The miRNA expression profiles of the four groups were compared by microarray analysis of plasma RNA extracts (Fig. S1A–D). The workflow of this study is shown in Fig. 1A. First, differentially expressed miRNAs with a fold-change ≥ 2 and P ≤ 0.05 were screened in the patients. Differentially expressed miRNAs in older adults were selected using the same analytical criteria. Finally, 51 candidates miRNAs common to both datasets were identified as age-associated miRNAs in patients with AAD, of which 36 were upregulated and 15 were downregulated (Fig. 1B). Further large-scale verification in the plasma confirmed the specific elevation of miR-1204 level in the older patient group (Fig. 1C), which positively correlated

with age (Fig. 1D). Remarkably, no significant difference was observed in the expression of miR-1204 between TAAD and AAA (Fig. 1E), suggesting generic age-associated upregulation in different aneurysmal diseases and AD. Human AAD and normal aortic tissue samples were obtained from patients or transplant donors who underwent surgery. Eight differentially expressed miRNAs were selected for verification using microarray analysis. Real-time quantitative reverse transcription polymerase chain reaction (qRT-PCR) showed that the levels of two miRNAs (miR-124-3p and miR-1204) were significantly different between the aortic tissues of healthy older participants and older patients with AAD (Fig. 1F). Notably, consistent with our findings, aberrant expression of miR-124-3p in AAD tissues has been reported previously[17]. Finally, in situ hybridization (ISH) showed that miR-1204 localized to the aortic media (Fig. 1G). Consistent with the plasma findings and as shown in Fig. 1G, the expression of miR-1204 was very low in the aortas of young healthy participants and slightly increased in the aortas of older healthy participants. However, the expression of miR-1204 was significantly increased in the aortas of young patients with AAD and dramatically enhanced in the aortas of older patients with AAD. These findings demonstrated increased miR-1204 expression in both the plasma and aorta of healthy participants and patients with AAD in an age-associated manner, suggesting that miR-1204 is an aging- and AAD-associated miRNA.

### miR-1204 aggravates angiotensinII (AngII)-induced AAD formation

To further explore the potential role of miR-1204 in AAD formation, a murine AAD model was established by an intravenous injection of miR-1204 agomir combined with 4-week AngII administration in C57BL/6 mice (Fig. 2A). We initially determined baseline expression levels of miR-1204 in the major organs of C57BL/6 mice. As shown in Fig. S2A, miR-1204 exhibited similar expression levels in various tissues, including the heart, lungs, kidneys, liver, and aorta. However, following AngII infusion, miR-1204 level was elevated in the plasma, peripheral blood mononuclear cells, and aorta compared to that in the control group (Fig. S2B–D). Next, we verified the target organs of the miR-1204 agomir using fluorescence imaging and found that the miR-1204 agomir accumulated in the heart, lungs, kidneys, liver, and aorta 6 h after tail vein injection (Fig. S2E). We found that the miR-1204 agomir did not affect the body weight or systolic blood pressure (SBP; Fig. S2F and Supplemental Table 2). Successful overexpression of miR-1204 in the mouse aorta was confirmed using qRT-PCR following five consecutive tail vein injections (Fig. S2G). Although the miR-1204 agomir did not affect AAD formation or survival, it markedly increased AngII-induced AAD incidence, leading to mouse death, a phenomenon not observed

## Table 1 | Clinical characteristics

| Characteristics | Young normal (70) | Young patient (116) | P value | Elder normal (88) | Elder patient (156) | P value |
|---|---|---|---|---|---|---|
| Age, y | 42.1 ± 5.6 | 40.3 ± 6.6 | 0.06 | 62.1 ± 4.5 | 64.2 ± 8.6 | 0.17 |
| Male, n (%) | 52 (74.2) | 98 (84.5) | 0.12 | 64 (72.7) | 122 (78.2) | 0.35 |
| Smoking, n (%) | 19 (27.1) | 42 (36.2) | 0.26 | 38 (43.1) | 71 (45.5) | 0.79 |
| Hypertension, n (%) | – | 75 (64.7) | – | – | 121 (77.6) | – |
| Diabetes mellitus, n (%) | – | 2 (1.7) | – | – | 15 (9.6) | – |
| CTD, n (%) | – | 8 (6.9) | – | – | 2 (1.3) | – |
| TG, mmol/L | 1.11 ± 0.48 | 1.25 ± 0.7 | 0.58 | 1.3 ± 0.75 | 1.28 ± 0.66 | 0.94 |
| TC, mmol/L | 3.72 ± 0.81 | 3.73 ± 0.52 | 0.98 | 3.7 ± 0.79 | 3.86 ± 0.44 | 0.51 |
| LDL, mmol/L | 2.38 ± 0.69 | 2.41 ± 0.42 | 0.91 | 2.36 ± 0.74 | 2.29 ± 0.32 | 0.76 |
| HDL, mmol/L | 1.14 ± 0.22 | 1.11 ± 0.22 | 0.78 | 1.08 ± 0.25 | 1.03 ± 0.16 | 0.47 |
| Uric acid, umol/L | 389.9 ± 156.7 | 359 ± 132.2 | 0.39 | 392.2 ± 153.3 | 346.1 ± 134.6 | 0.16 |

Data are presented as means ± standard deviation or the number of patients (n, %). The unpaired two-tailed Student's t test or the Mann–Whitney U test was applied to evaluate statistical significance between continuous variables with or without normal distribution, respectively. The Chi-square test was used to evaluate the statistical significance between the proportions of the two groups.
CTD connective tissue diseases, TG triglyceride, TC total cholesterol, LDL low-density lipoprotein, HDL high-density lipoprotein.

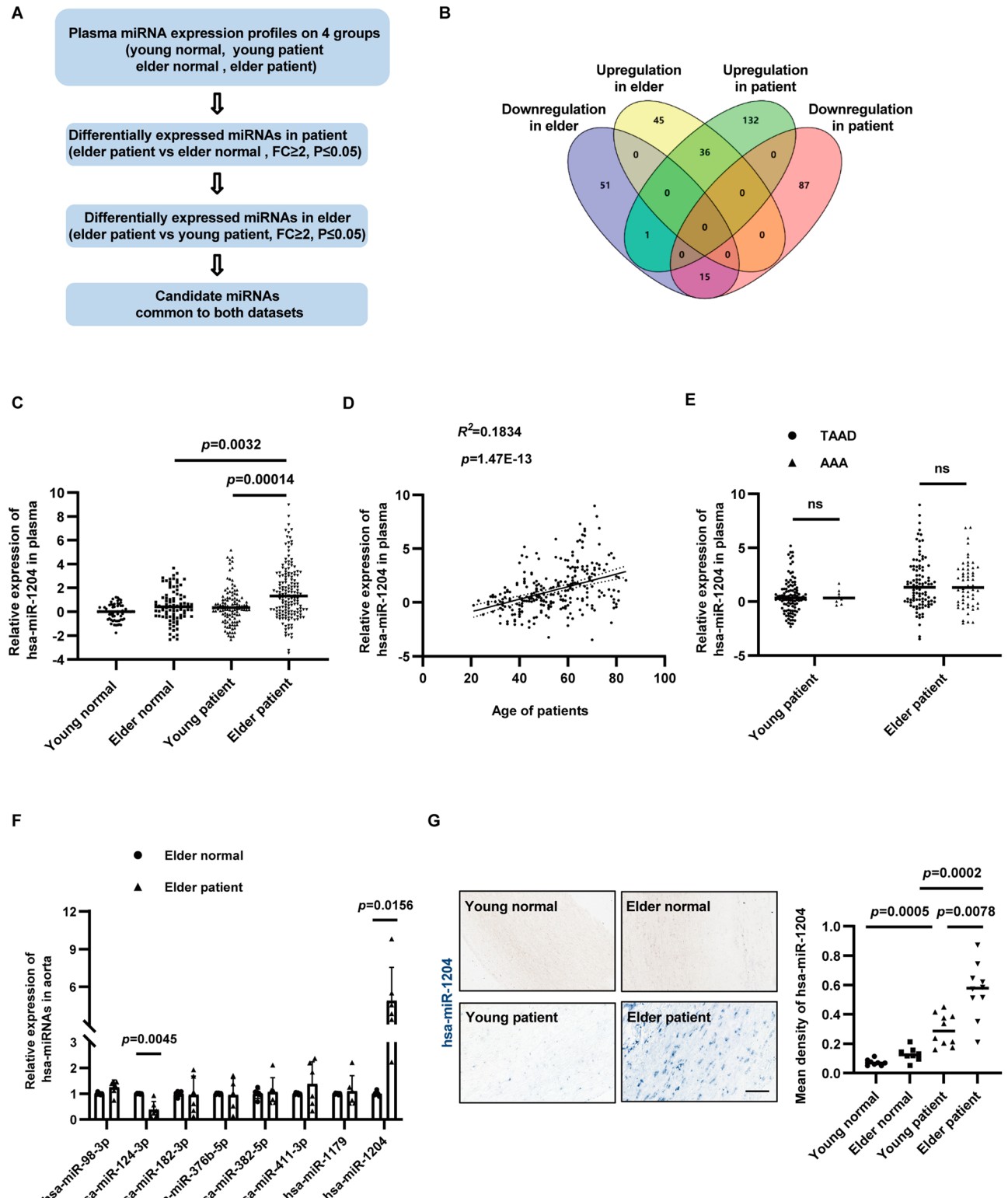

in the group receiving AngII alone (Fig. 2B–D). Furthermore, ultrasonography revealed dilation of the aorta in the miR-1204 agomir + AngII group compared to that in the AngII group (Fig. 2E–I). Additionally, hematoxylin and eosin (H&E) and elastic van Gieson (EVG) staining showed that miR-1204 agomir exacerbated AngII-induced AAD pathological progress, including intima tears, pseudolumen formation, and breakdown of elastic layers (Fig. 2J). Additionally, miR-1204 agomir significantly increased senescence-associated-β-galactosidase (SA-β-gal) expression in the vascular walls of AngII-infused mice, indicating a

strong effect of miR-1204 in inducing cell senescence (Fig. 2J). These findings indicate that miR-1204 is a crucial mediator of Ang II-induced AAD formation.

## Inhibition of miR-1204 attenuates β-aminopropionitrile mono-fumarate (BAPN) induced AAD formation

Next, we established another model in which AAD was induced by BAPN administration to estimate the possible beneficial effects of miR-1204 deficiency on AAD formation. Locked nucleic acid (LNA) anti-miR-

**Fig. 1 | miR-1204 is upregulated with aging in patients with aortic aneurysm and dissection (AAD). A** miRNA-selective strategies in plasma samples. **B** Venn diagram showing common miRNAs in older adults and patient datasets. **C** Plasma miR-1204 expression was determined by real-time quantitative reverse transcription-polymerase chain reaction (qRT-PCR) in the indicated groups. $n = 70$ (young normal), $n = 88$ (elder normal), $n = 116$ (young patient), and $n = 156$ (elder patient). Statistical analysis was performed using Kruskal–Wallis test with Dunn's multiple comparison test. **D** Correlation between miR-1204 expression in the plasma and age of patients with AAD. $n = 272$. Statistical analysis was performed using two-tailed Spearman's correlation test. **E** Differences in plasma miR-1204 expression between patients with thoracic aortic aneurysm and dissection (TAAD) and those with abdominal aortic aneurysm (AAA) were determined using qRT-PCR. $n = 108$ (young patients with TAAD), $n = 8$ (young patients with AAA), $n = 101$ (elder patients with TAAD), and $n = 55$ (elder patients with AAA). Statistical analysis was performed using two-tailed Mann–Whitney $U$ test. **F** Expression of several differentially expressed miRNAs (identified in panel **B**) in aortic samples from healthy participants and patients with AAD were determined using qRT-PCR. $n = 6$ except for hsa-miR-376b-5p in elder patient group, $n = 5$ for hsa-miR-376b-5p in elder patient group. Statistical analysis was performed using two-tailed Student's $t$ test with Welch's correction. **G** Representative images (left) and quantification (right) of miR-1204 in the aortas of healthy participants and patients with AAD detected by in situ hybridization staining. $n = 8$ (young normal), $n = 8$ (elder normal), $n = 10$ (young patient), $n = 10$ (elder patient). Statistical analysis was performed using two-tailed Welch's ANOVA followed by Dunn's multiple comparison test. Scale bar indicates 100 μm. Data are presented as mean ± SD. ns not significant. Source data are provided as a source data file.

1204 or LNA scrambled-miR (LNA scr-miR) was delivered via tail vein injection into C57BL/6 mice prior to 4-week BAPN administration (Fig. 3A). Successful inhibition of miR-1204 expression in the mouse aorta was confirmed by qRT-PCR following the tail vein injection (Fig. S2H). We found that LNA anti-miR-1204 did not affect body weight (Supplemental Table 3). As shown in Fig. 3B–D, anti-miR-1204 treatment reduced the incidence and mortality of AAD. The aortic arch diameter in miR-1204 deficient mice was smaller than that in scrambled miR mice (Fig. 3E, F). Media dissection, pseudolumen generation, and elastic layer breakage in miR-1204 deficient mice were less severe, as determined by H&E and EVG staining (Fig. 3G). SA-β-gal staining showed that miR-1204 deficiency reduced vascular cell senescence (Fig. 3H). These findings indicated that miR-1204 inhibition ameliorates vascular cell senescence and AAD formation.

### miR-1204 promotes SASP components accumulation and contractile phenotype loss in VSMCs

Given the increased expression of miR-1204 in the aortic media of elder patients and its important role in vascular cell senescence and AAD formation, we sought to elucidate the underlying molecular mechanisms of miR-1204 in VSMCs. Primary VSMCs were transfected with a miR-1204 mimic and subjected to tandem mass tag (TMT)-based quantitative proteomic analysis. Using an adjusted $P < 0.05$ as the significance threshold and 1.3 as the cutoff ratio for fold-change, 77 differentially expressed proteins were identified in miR-1204 overexpressing VSMCs. Of these, 33 were upregulated and 44 were downregulated (Fig. S3A–C). Differentially expressed proteins were subjected to enrichment analysis using the kyoto encyclopedia of genes and genomes (KEGG) pathway and gene ontology (GO) classification. In the KEGG pathway analyses, the differentially expressed proteins were primarily related to cellular senescence, cytokine-cytokine receptor interactions, and cell cycle (Fig. S3D). GO analysis also revealed that most of these proteins were involved in the development of the cardiovascular system (GO: 0072358) and regulation of programmed cell death (GO:0043068) (Fig. S3E). These bioinformatic analyses annotated the role of miR-1204 in VSMC senescence.

We hypothesized that miR-1204 plays a detrimental role by regulating VSMC senescence. First, a significant accumulation of SA-β-gal positive cells was observed in VSMCs overexpressing miR-1204 (Fig. 4A). Consistently, several SASP components including interleukin 6 (IL-6), IL-8, monocyte chemotactic protein 1 (MCP-1), CXC chemokine ligand 1 (CXCL1), CXC chemokine ligand 2 (CXCL2), and insulin-like growth factor-binding protein1 (IGFBP3) were upregulated at the transcriptional level in VSMCs overexpressing miR-1204 (Fig. 4B). Furthermore, miR-1204 overexpression upregulated the expression of IL-6 and MCP-1 in the supernatants of cultured VSMCs (Fig. 4C, D). Notably, the combined application of miR-1204 and AngII showed a synergistic effect on VSMC inflammatory responses in vitro. As major components of the arterial wall, the fate of VSMCs substantially influences the development of arterial diseases[18]. Several studies have highlighted the importance of phenotypic changes in VSMC in AAD formation[19,20]. Because VSMCs transfected with miR-1204 acquired SASP properties, we further investigated whether the accumulation of these SASP components facilitate VSMC dedifferentiation. Immunoblotting analysis of VSMC contractile markers, including α-smooth muscle actin (α-SMA), smooth muscle protein 22 (SM22), and myosin heavy chain 11 (MYH11), revealed that miR-1204 overexpression reduced the expression of contractile markers in VSMC (Fig. 4E–G).

To further validate the cellular level (in vitro) results, we determined the role of miR-1204 in the induction of VSMC dedifferentiation in vivo. The miR-1204 agomir significantly increased CD68 expression in the adventitial layer of AngII-infused mice, suggesting that miR-1204 strongly induced inflammatory infiltration (Fig. 4H). Furthermore, the expression of VSMC contractile markers α-SMA, SM22, and MYH11 was reduced in the aortic media following AngII treatment, the effects of which were amplified by miR-1204 agomir (Fig. 4H). These results were confirmed by immunofluorescence (IF) staining of patient samples. VSMC contractile markers were lost in an age-dependent manner, together with the apparent accumulation of SASP components in the lesions (Fig. S4A–J). Collectively, these data demonstrate that miR-1204 promotes SASP component accumulation and contractile phenotype loss in VSMCs, both in vitro and in vivo.

### MYLK is a direct target of miR-1204 in VSMCs

MiRNAs negatively regulate gene expression by binding to corresponding messenger RNAs (mRNAs), thereby causing degradation or translation inhibition[21]. To identify the direct targets that mediate miR-1204-induced pathological effects, we used bioinformatics analyses (TargetScan) for target prediction. A total of 112 candidate genes related to blood vessels were identified, including nine VSMC-related genes (Supplemental Table 4). Next, we compared the mRNA expression of these genes in miR-1204 overexpressing VSMCs with that to in control cells (Fig. 5A). Among the potential targets, the expression of three genes were significantly inhibited (adjusted $P < 0.05$, fold change > 2), and MYLK was the only gene associated with VSMCs. Interestingly, AngII administration enhanced the miR-1204-mediated inhibition of potential targets, indirectly suggesting that AngII stimulation activates miR-1204 expression in VSMCs (Fig. 5B). We found that the miR-1204 mimic significantly decreased MYLK expression in VSMCs (Fig. 5C). IF staining revealed that the miR-1204 agomir decreased MYLK expression in the aorta, and its function was augmented by AngII co-administration (Fig. 5D). Based on the presence of a complementary miR-1204 binding sequence in the MYLK 3' untranslated region (3' UTR), we validated this hypothesis using 3'UTR luciferase assays. We overexpressed miR-1204 in HEK-293T cells transfected with luciferase expression plasmids containing the wild-type, mutated sequence, or deletion sequence of MYLK 3' UTR. Increased miR-1204 expression resulted in decreased luciferase activity, whereas mutation or deletion of the target sequence restored luciferase activity (Fig. 5E). Finally, we verified whether MYLK expression was decreased in the aortas of patients with AAD. Coincidentally, abundant MYLK expression was

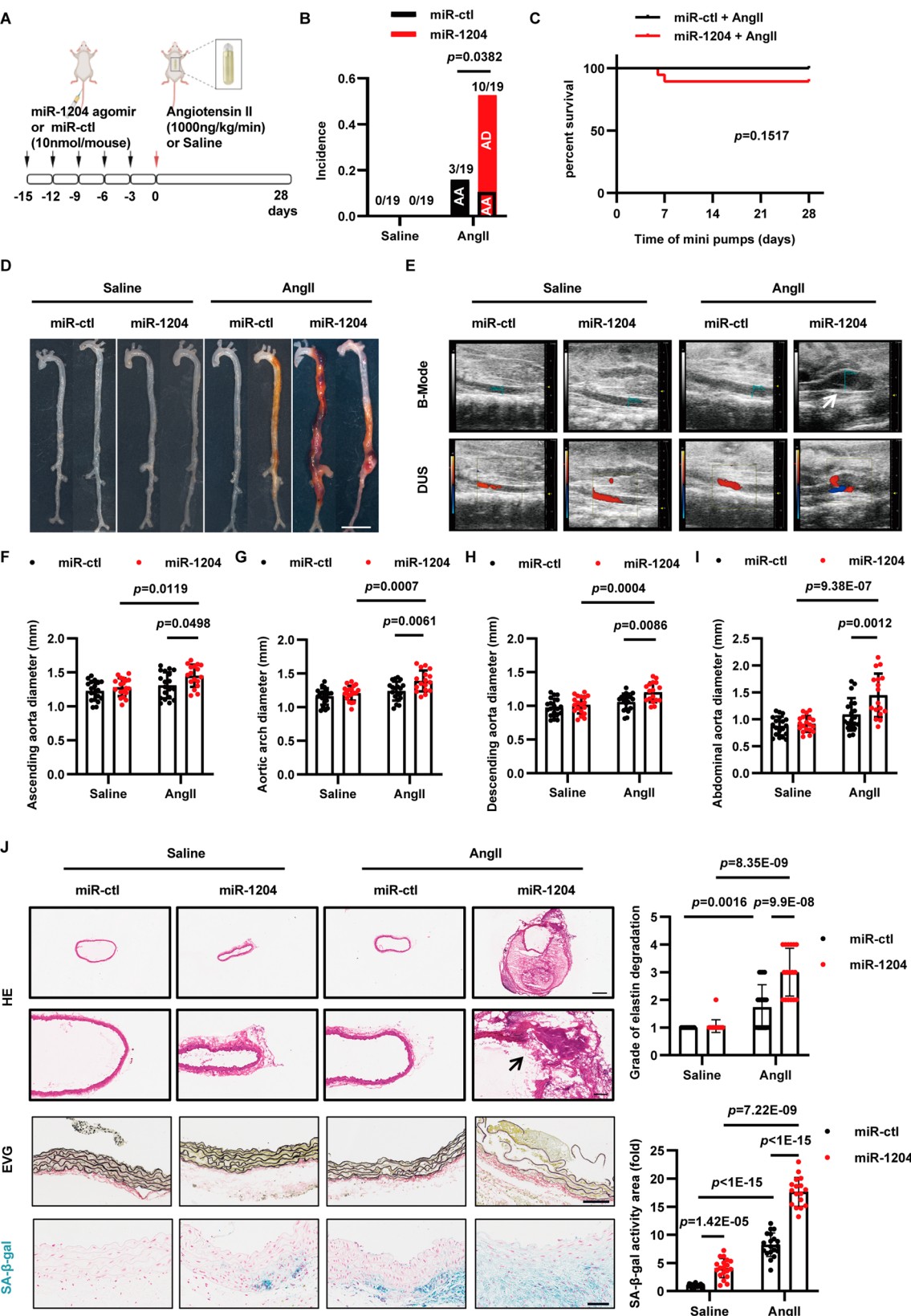

observed in the aortas of healthy young participants. The expression of MYLK was significantly decreased in the aortas of healthy older participants compared to that of healthy young participants. More importantly, the expression of MYLK was reduced in the aorta of young patients with AAD compared to that in young healthy participants and was reduced even more in the aorta of older patients with AAD

(Fig. 5F). Taken together, these findings strongly indicated that miR-1204 directly regulates MYLK expression.

### MYLK modulates SASP component accumulation in VSMCs
Given the considerable effect of miR-1204 on SASP, we assessed whether silencing of MYLK has a similar effect in VSMCs and whether

**Fig. 2 | miR-1204 aggravates angiotensin II (AngII) induced aortic aneurysm and dissection (AAD) formation. A** Scheme of AngII-induced AAD experiments. MiR-1204 agomir or miR-control (miR-ctl) (10 nmol) was administered to 4-month-old male C57BL/6 mice by tail vein injection every 3 d for a total of five times before 4-week AngII-infusion. **B** AAD incidence. AA aortic aneurysm, AD aortic dissection. Statistical analysis was performed using two-tailed Fisher's exact test. **C** Kaplan−Meier curves of survival in the indicated groups. Statistical analysis was performed using log-rank test. **D** Representative photographs of aortas in the indicated groups. Bar indicates 4 mm. **E** Representative B-mode ultrasound and doppler ultrasound (DUS) detection of abdominal aortas. The white arrow indicates an aneurysm. **F−I** Quantification of ascending aorta, aortic arch, descending aorta and abdominal aorta diameters using ultrasound. $n = 17$ biological replicates for miR-1204 + AngII group. $n = 19$ biological replicates for other groups. Statistical analysis was performed using a one-way ANOVA followed by Tukey's multiple comparison test. **J** Representative images of hematoxylin & eosin (H&E) staining of the abdominal aortas. The black arrow indicates an intimal tear. Upper bar indicates 400 μm. Lower bar indicates 100 μm. Representative images of elastic van Gieson (EVG) staining (left) and elastin degradation grade (right) in the abdominal aorta. $n = 17$ biological replicates for miR-1204 + AngII group. $n = 19$ biological replicates for other groups. Statistical analysis was performed using the Kruskal−Wallis test with Dunn's multiple comparisons test. Scale bar, 100 μm. Representative images of senescence-associated-β-galactosidase (SA-β-gal) staining (left) and quantification of SA-β-gal activity area (right) in the abdominal aorta. $n = 17$ biological replicates for miR-1204 + AngII group. $n = 19$ biological replicates for other groups. Statistical analysis was performed using two-tailed Welch's ANOVA followed by Dunn's multiple comparison test. Scale bar indicates 50 μm. Data are presented as mean ± SD. Source data are provided as a source data file.

MYLK overexpression rescues the AngII-induced inflammatory damage. First, the overexpression efficacy of the MYLK adenovirus was verified. VSMCs were infected with adenovirus particles at various multiplicity of infection (MOI), and MYLK mRNA and protein levels were assessed. As shown in Fig. S5A, B, MYLK expression levels were significantly increased at an MOI of 10. The protein expression levels of IL-6 and MCP-1 were confirmed using an enzyme-linked immunosorbent assay (ELISA). As expected, MYLK silencing augmented AngII-induced IL-6 and MCP-1 expression in VSMCs, whereas MYLK overexpression mitigated these effects (Fig. S6A−D). Subsequent qRT-PCR analyses of SASP components including IL-6, MCP-1, tumor necrosis factor-α, and MMP9 validated our findings at the transcriptional level (Fig. S6E−L). Gelatin zymography showed that matrix metalloproteinase (MMP) activity in the culture medium of AngII-treated VSMCs was also elevated following MYLK silencing, whereas it was attenuated following MYLK overexpression (Fig. S6M, N).

### MYLK ameliorates miR-1204-induced SASP and contractile phenotype changes both in vitro and in vivo

To gain further insight into the role of MYLK in restoring miR-1204-mediated VSMC damage, we conducted a series of rescue experiments by overexpressing MYLK. MYLK overexpression markedly alleviated SA-β-gal accumulation in VSMCs following miR-1204 transfection and AngII application (Fig. 6A). IL-6 and MCP-1 protein levels in the supernatant, as quantified by ELISA were significantly reduced after MYLK overexpression in VSMCs transfected with the miR-1204 mimic, with or without AngII treatment (Fig. 6B, C). Accordingly, the mRNA expression levels of IL-6, MCP-1, and several SASP components were significantly decreased by MYLK overexpression in VSMCs transfected with the miR-1204 mimic (Fig. 6D). In contrast, MYLK overexpression enhanced the expression of contractile markers and attenuated the ability of miR-1204 to inhibit these contractile markers in VSMCs, which were more obvious following AngII stimulation (Fig. 6E−G). These findings support the notion that the effect of miR-1204 on VSMC senescence and phenotypic switching is a result of the direct regulation of MYLK.

To further validate the MYLK-mediated rescue of miR-1204-induced AAD and determine whether this effect is age-related, we conducted additional in vivo experiments using the AngII-induced AAD model (Fig. 6H). The miR-1204 agomir + AngII group showed a substantially higher incidence of AAD and SA-β-gal positive staining in contrast with control groups. As expected, the MYLK overexpression group exhibited a significant decrease in AAD incidence. Notably, SA-β-gal staining was also markedly suppressed (Fig. 6I−K). The abdominal aorta diameter in mice from the miR-1204 + AngII group was significantly larger than that in control mice, whereas MYLK overexpression rescued aortic dilation (Fig. 6L). IF staining confirmed effective overexpression of MYLK in the aorta. Additionally, the expression of SASP markers including IL-6 and MCP-1 was increased in the aortic media of the miR-1204 + AngII group and these effects were attenuated by MYLK overexpression (Fig. 6M). Finally, the contraction of aortic rings isolated from AngII-treated mice in response to the vasoconstrictor phenylephrine was measured ex vivo. Aortas overexpressing miR-1204 exhibited reduced contractility following phenylephrine stimulation. In contrast, MYLK overexpression significantly alleviated the miR-1204-induced contractile strength reduction (Fig. 6N). These data provided direct evidence that miR-1204 and MYLK in VSMCs contribute to AAD through an age-related mechanism.

### MYLK ameliorates the effect of miR-1204 on macrophages and VSMC properties

We established a Transwell culture system incorporating PMA-differentiated THP-1 macrophages grown on the insert membrane, with conditioned medium from miR-1204 overexpressing VSMCs added to the wells (Fig. S7A). The conditioned medium significantly induced macrophage migration in the Transwell chambers. However, MYLK overexpression markedly inhibited this effect (Fig. S7B). In addition, we validated the effect of conditioned medium treatment on the macrophage phenotype. M1-type macrophages promote inflammatory processes. Therefore, levels of CD80, a representative marker of the M1 phenotype, was detected using flow cytometry. Compared with the control group cells, the conditioned medium significantly induced macrophages to polarize toward the M1 phenotype. MYLK overexpression attenuated this effect (Fig. S7C). These results suggest that overexpression of miR-1204 induces the secretion of cytokines by VSMCs, fostering macrophage migration and M1 polarization. Notably, this process is regulated by MYLK.

Consistent with the miR-1204-mediated effects on VSMC senescence, overexpression of miR-1204 in VSMCs resulted in a notable suppression of cell proliferation, as determined by an 5-Ethynyl-2′-deoxyuridine (EdU) proliferation assay (Fig. S7D). MYLK overexpression attenuated miR-1204-induced on VSMC proliferation. Furthermore, VSMCs overexpressing miR-1204 exhibited reduced contractility, as evidenced by collagen gel contraction assay. MYLK overexpression abrogated the effects of miR-1204 on VSMC contractility (Fig. S7E). Moreover, VSMCs overexpressing miR-1204 exhibited reduced migration as indicated by a scratch-wound assay. MYLK overexpression attenuated the inhibitory effect of miR-1204 on VSMC migration (Fig. S7F). Taken together, these findings indicate that miR-1204 inhibits VSMC proliferation, migration, and contractility by regulating MYLK expression.

### MYLK ameliorates miR-1204-induced phenotype switch via transforming growth factor-β (TGF-β) signaling pathway in VSMCs

The TGF-β signaling pathway exerts protective effects by inducing contractile protein expression and inhibiting inflammation in the aortic wall[22]. Therefore, we hypothesized that MYLK may rescue miR-1204-induced damage through the TGF-β signaling pathway. As shown

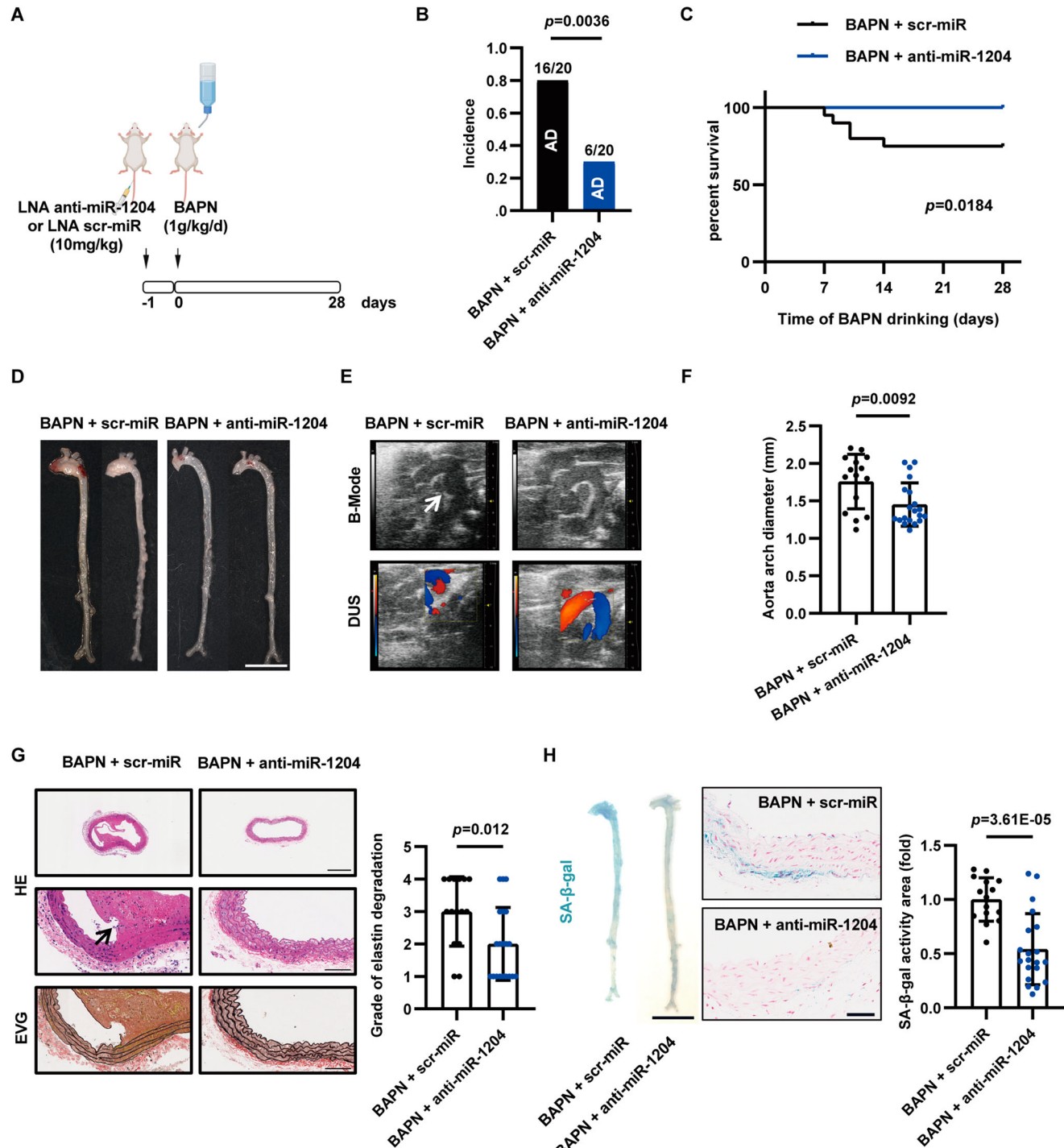

**Fig. 3 | Inhibition of miR-1204 ameliorates β-aminopropionitrile mono-fumarate (BAPN)-induced aortic aneurysm and dissection (AAD) formation.**
**A** Scheme of BAPN-induced AAD experiments. LNA anti-miR-1204 or LNA scramble-miR (LNAscr-miR) were administered (10 mg/kg) via tail vein injection to 3-week-old male C57BL/6 mice prior to administration of 1 g/kg/day BAPN in drinking water for four weeks. **B** AAD incidence. AD: aortic dissection. Statistical analysis was performed using two-tailed Fisher's exact test. **C** Kaplan–Meier curves of survival in the indicated groups. Statistical analysis was performed using log-rank test.
**D** Representative photographs of aortas in the indicated groups. Scale bar represents 4 mm. **E** Representative B-mode ultrasound and doppler ultrasound (DUS) detection of the aortic arch. The white arrow indicates an aneurysm.
**F** Quantification of aortic arch diameters measured by ultrasound. $n = 15$ biological replicates for BAPN + scr-miR group. $n = 20$ biological replicates for BAPN +

anti-miR-1204. Statistical analysis was performed using two-tailed Student's $t$ test.
**G** Representative images of hematoxylin & eosin (H&E) staining of the aortic arch. The black arrow indicates an intimal tear. Upper scale bar, 400 μm. Lower scale bar, 100 μm. Representative images of elastic van Gieson (EVG) staining (left) and elastin degradation grade (right) in the aortic arch. $n = 15$ biological replicates for BAPN + scr-miR group. $n = 20$ biological replicates for BAPN + anti-miR-1204. Statistical analysis was performed using two-tailed Mann–Whitney $U$ test. Scale bar, 100 μm. **H** Representative images of senescence-associated-β-galactosidase (SA-β-gal) staining (left) and quantification of the SA-β-gal activity area (right) in the aortic arch. $n = 15$ biological replicates for BAPN + scr-miR group. $n = 20$ biological replicates for BAPN + anti-miR-1204. Statistical analysis was performed using two-tailed Student's $t$ test. Left bar represents 4 mm. Right bar indicates 50 μm. Data were presented as mean ± SD. Source data are provided as a source data file.

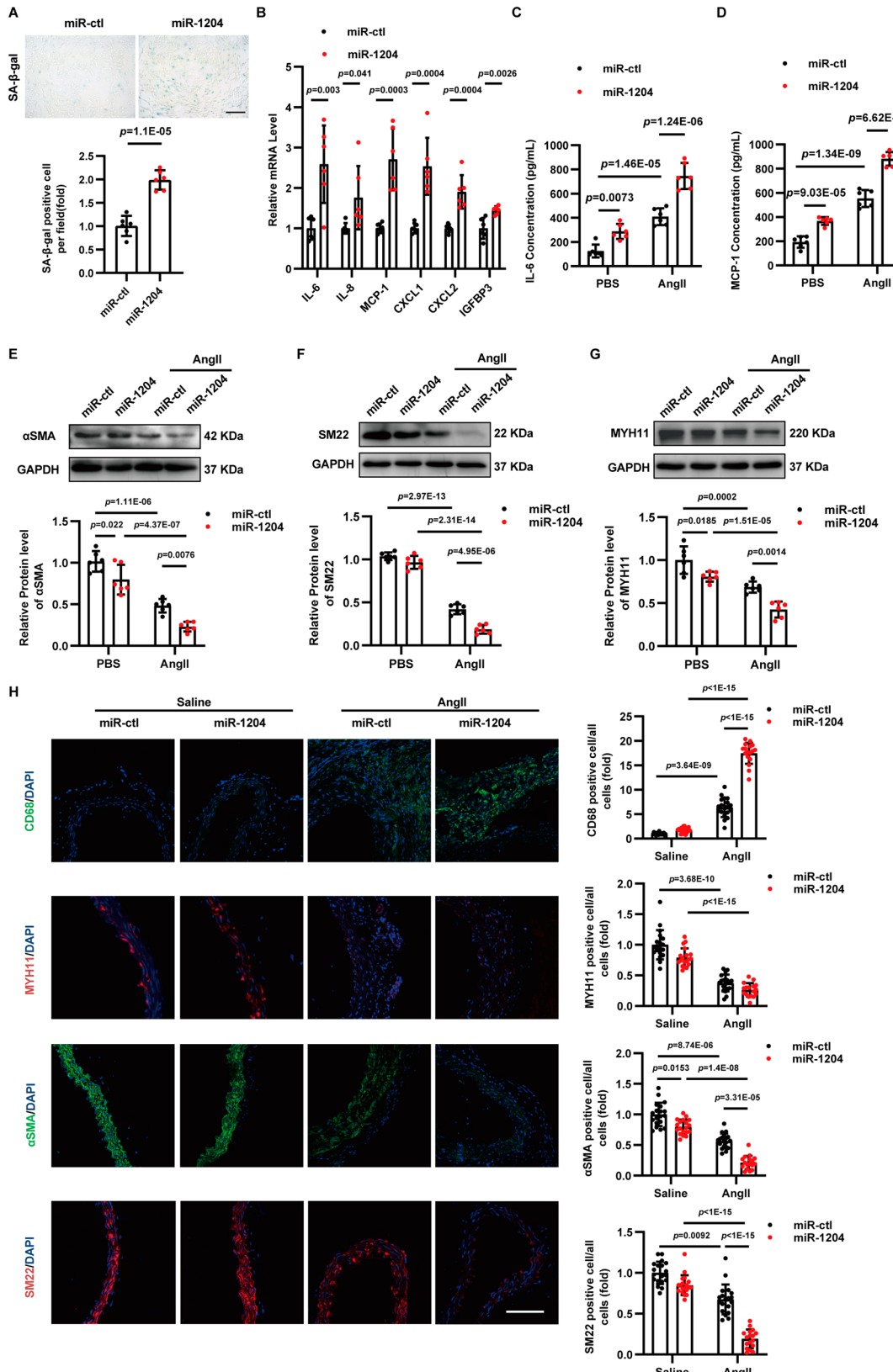

in Fig. S8A, B, miR-1204 reduced the protein expression of TGFβR2 and SMAD2/3 in the cultured VSMCs. However, overexpression of MYLK increased the expression of TGFβR2 and SMAD2/3 in VSMCs, with or without AngII treatment (Fig. S8C, D). Furthermore, MYLK reversed miR-1204-mediated inhibition of TGFβR2 and SMAD2/3 in VSMCs, even post AngII treatment (Fig. S8E, F).

## External factors that induce cell senescence upregulate miR-1204 through p53 interaction with PVT1 in VSMCs

Next, we compared the basal expression levels of miR-1204 with those of other benchmark miRNAs in VSMCs. Endogenous expression of miR-1204 was lower than that of the miR143/145 cluster (Fig. S9A). Mature miR-1204 expression was significantly downregulated by TGF-

**Fig. 4 | miR-1204 promotes senescence-associated secretory phenotype (SASP) component accumulation and contractile phenotype loss in vascular smooth muscle cells (VSMCs). A** Representative images of senescence-associated-β-galactosidase (SA-β-gal) staining (upper panel) and quantification (lower panel) in VSMCs transfected with miR-1204 mimics or miR-control (miR-ctl). $n = 6$ biological replicates. Statistical analysis was performed using two-tailed Student's $t$ test. Scale bar, 100 μm. **B** Quantification of mRNA levels of SASP components in VSMCs transfected with miR-1204 mimics or miR-ctl. $n = 6$ biological replicates. Statistical analysis was performed using two-tailed Student's $t$ test. **C** Interleukin 6 (IL-6) concentration in the culture supernatant of VSMCs transfected with miR-1204 mimics or miR-ctl combined with angiotensin II (AngII) treatment. $n = 6$ biological replicates. Statistical analysis was performed using a one-way ANOVA followed by Tukey's multiple comparison test. **D** Monocyte chemotactic protein 1 (MCP-1) concentration in the culture supernatant of VSMCs transfected with miR-1204 mimics or miR-ctl combined with AngII treatment. $n = 6$ biological replicates.

Statistical analysis was performed using a one-way ANOVA followed by Tukey's multiple comparison test. **E−G** Representative plots (upper) and quantification (lower) of immunoblot analysis of contractile markers, α-smooth muscle actin (α-SMA), smooth muscle protein 22 (SM22), and myosin heavy chain 11 (MYH11) in VSMCs transfected with miR-1204 mimics or miR-ctl combined with AngII treatment. $n = 6$ biological replicates. Statistical analysis was performed using a one-way ANOVA followed by Tukey's multiple comparison test. **H** Representative images (left) and quantification (right) of immunofluorescence staining of CD68, MYH11, α-SMA, and SM22 in mouse aortas in the indicated groups. $n = 17$ biological replicates for miR-1204 + AngII group. $n = 19$ biological replicates for other groups. Statistical analysis was performed using a one-way ANOVA followed by Tukey's multiple comparison test for α-SMA and SM22 quantification, by two-tailed Welch's ANOVA followed by Dunn's multiple comparisons for CD68 and MYH11 quantification. Scale bar, 100 μm. Data are presented as mean ± SD. Source data are provided as a source data file.

---

β (Fig. S9B) but upregulated by PDGF-BB (Fig. S9C), suggesting that miR-1204 was transcriptionally regulated during VSMC phenotypic modulation. Interestingly, the effect of PDGF-BB on miR-1204 expression was marginal, indicating that the proliferative pathway may not govern miR-1204 induction in VSMCs. Given that plasma miR-1204 levels in patients were increased and positively correlated with their age, we sought to clarify the transcriptional process of miR-1204 in senescent cells. In vitro, cell senescence is triggered in response to various stresses such as oxidative stress, radiation, or chemical agents that cause DNA damage[23]. Therefore, we examined pri-mir-1204 expression in VSMCs under different cell stresses, including AngII, hydrogen peroxide ($H_2O_2$), and daunorubicin, and found that pri-mir-1204 expression increased in VSMCs (Fig. 7A). Similar to pri-mir-1204 expression, mature miR-1204 expression also increased significantly in cells treated with AngII compared with that in untreated control cells (Fig. S9D).

A previous study reported that miR-1204 is encoded by the PVT1 locus on human chromosome 8q24. The tumor suppressor protein p53 binds to and activates a canonical response element within the locus, thereby increasing miR-1204 expression in colorectal carcinoma cells[24]. Owing to the central role of p53 stabilization and accumulation in cell senescence, we hypothesized that miR-1204 is upregulated in VSMCs via a p53-dependent mechanism during aging. First, we measured the expression of pri-mir-1204 in p53-activated VSMCs. As expected, nutlin-3a treatment led to p53 activation and pri-mir-1204 induction. However, p53 silencing eliminated this effect (Fig. 7B). Consistent with the changes in the expression of pri-mir-1204, p53 activation induced miR-1204 expression, and p53 silencing abrogated this effect (Fig. S9E). Chromatin immunoprecipitation (ChIP) experiments corroborated these findings (Fig. 7C). The binding of p53 to the PVT1 response element (PVT1 RE) was enhanced in daunorubicin-treated VSMCs, with its affinity reaching a comparable degree to that in the binding between p53 and its classic binding site, the p21 response element. Next, a luciferase assay was performed to confirm the interaction between p53 and the PVT1 RE. VSMCs were transfected with pCDNA3.1(+) plasmid, clones containing the p53 domain, and pGL3-basic plasmid clones containing PVT1 RE. Increased luciferase activity revealed that the p53 domain only binds to the PVT1 RE-containing plasmid and not the mutant or blank plasmids (Fig. 7D).

Having determined that miR-1204 is induced in senescent VSMCs in a p53-dependent manner, we tested whether miR-1204 influences p53 expression. In contrast to the results of a previous study, overexpression of miR-1204 increased p21, p16, and γ-H2AX protein levels and not that of p53 in VSMCs (Fig. 7E, F). Conversely, MYLK overexpression inhibited the expression of these genes without affecting the expression levels of p53. Moreover, miR-1204 markedly inhibited the cell division of the cultured VSMCs. Increased cell division was further confirmed by MYLK overexpression (Fig. 7G, H).

In addition to the regulation of its p53 expression, multiple molecular mechanisms govern p53 regulation. Therefore, ChIP-Seq assays were performed using p53 antibodies in the VSMCs following miR-1204 overexpression. KEGG analysis indicated that many target genes binding to p53 were enriched in pathways related to cell senescence and cell cycle in VSMCs overexpressing miR-1204 (Fig. 7I). We compared the peak distributions of SASP marker genes, including MCP-1, IL-8, CXCL1, and IGFBP3, in control and miR-1204 overexpressing cells, and found that miR-1204 overexpression markedly increased the association between p53 and SASP marker genes (Fig. 7J). The induction and activation of p53 mainly involve the uncoupling of p53 from its negative regulators, principally MDM2. Therefore, we examined the phosphorylation of MDM2 and the association between MDM2 and p53 in cultured cells. As expected, the phosphorylation of MDM2 was inhibited and, furthermore, its binding to p53 was attenuated in miR-1204 overexpressing VSMCs. In contrast, MYLK overexpression increased the MDM2/p53 interaction and MDM2 phosphorylation (Fig. 7K, L). These observations suggest that miR-1204, a transcriptional target of p53, regulates p53 activity through a feedback mechanism, which involves modulation of the interaction between p53 and MDM2.

Together, these findings support our hypothesis that miR-1204 is induced in senescent VSMCs in a p53-dependent manner and that miR-1204 induces VSMC senescence by regulating the activity of p53 and intervening with multiple markers of cellular senescence through a positive feedback loop.

## Discussion

In the present study, using both AngII and BAPN animal models, we demonstrated that aging aggravates AAD formation via the miR-1204-MYLK signaling axis. The present study contributes several novel insights to the existing literature. First, miR-1204 aggravates AAD formation, and inhibition of miR-1204 attenuates AAD formation. Second, aging induces miR-1204 expression via p53 interaction with PVT1, and miR-1204, in turn, induces cell senescence to form a positive feedback loop. Third, miR-1204 directly targets MYLK to induce VSMCs to acquire the SASP and lose their contractile phenotype, leading to cytokine/chemokine release, vascular inflammatory infiltration, and VSMC dedifferentiation.

Although TAA, AAA, and AD are aortic diseases, they are recognized as distinct ailments[25]. They exhibit several distinct differences, including differences in population prevalence, modes of inheritance, predisposing genes, and environmental risk factors. Genetic predisposition is more common in the occurrence of TAA and AD and includes multifaceted syndromes such as Marfan, Loeys–Dietz, type IV Ehlers–Danlos, and autosomal-dominant familial patterns of inheritance, whereas environmental risk factors are more common in AAA. The embryologic origins of the aorta (the ascending and arch aorta from the neural crest and the descending and abdominal aorta from

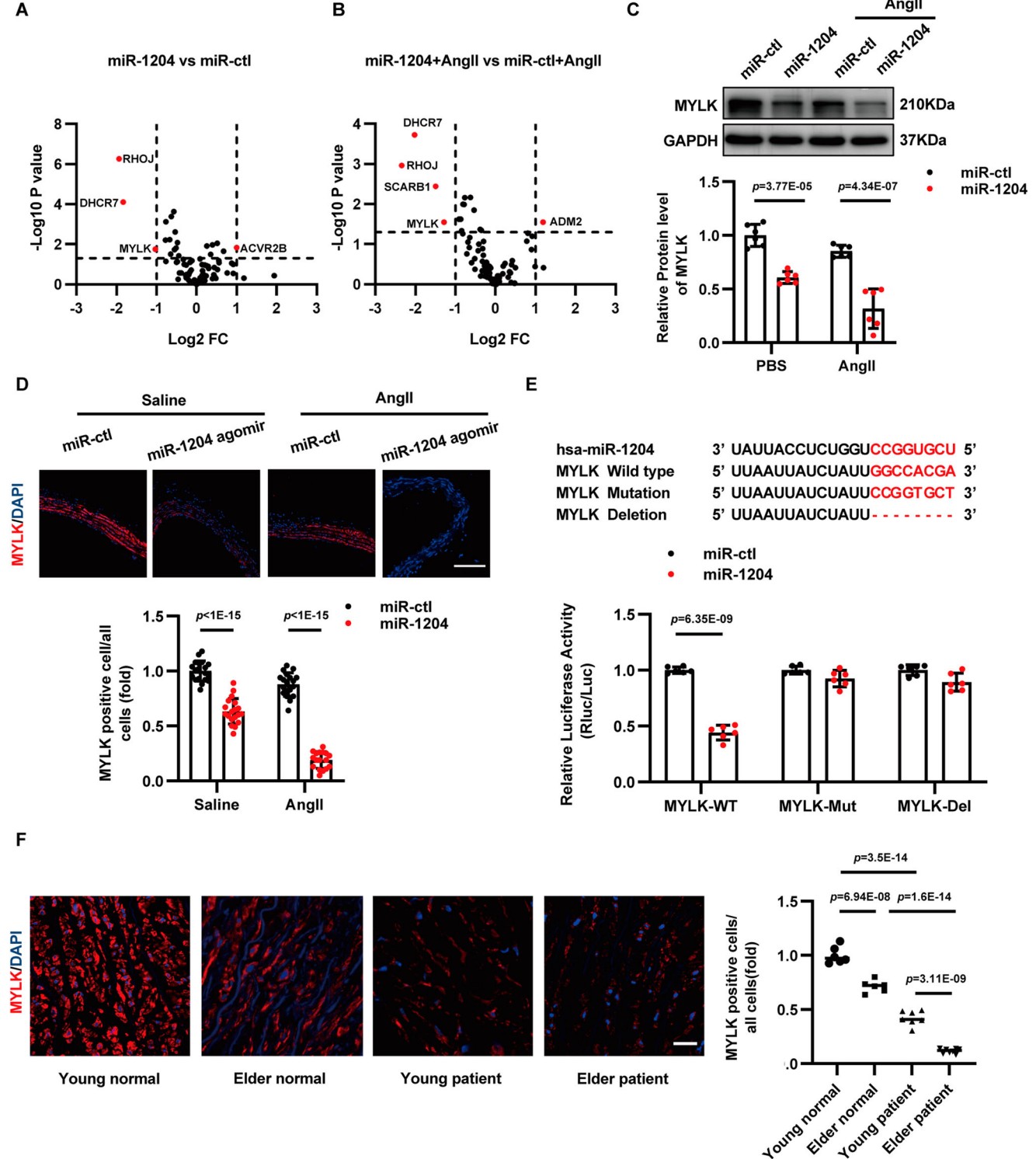

**Fig. 5 | Myosin light chain kinase (MYLK) is a direct target of miR-1204 in vascular smooth muscle cells (VSMCs). A**, **B** Volcano plot showing the mRNA expression of potential miR-1204 targets in VSMCs transfected with miR-1204 mimic or miR-control (miR-ctl) combined with angiotensin II (AngII) treatment (**B**) or not (**A**). *n* = 3 biological replicates. Statistical analysis was performed using a two-tailed Student's *t* test. **C** Representative plots (upper) and quantification (lower) of MLYK immunoblot analysis in VSMCs transfected with miR-1204 mimic or miR-ctl combined with AngII treatment. *n* = 6 biological replicates. Statistical analysis was performed using a one-way ANOVA followed by Tukey's multiple comparison test. **D** Representative images (upper panel) and quantification (lower panel) of MYLK immunofluorescence staining of mouse aortas from in the indicated groups. *n* = 17 biological replicates for the miR-1204 + AngII group. *n* = 19 biological replicates for

other groups. Statistical analysis was performed using a one-way ANOVA followed by Tukey's multiple comparison test. Scale bar represents 100 μm. **E** Potential hsa-miR-1204 binding site in MYLK (upper panel). Luciferase activity assay confirmed that MYLK is a direct target of miR-1204 (lower panel). *n* = 6 biological replicates. Statistical analyses were performed using two-tailed Student's *t* tests.
**F** Representative images (left) and quantification (right) of MYLK immunofluorescent staining in aortas of healthy participants and patients with AAD. *n* = 6 (young normal), *n* = 6 (elder normal), *n* = 7 (young patient), *n* = 8 (elder patient). Statistical analysis was performed using a one-way ANOVA followed by Tukey's multiple comparison test. Scale bar represents 20 μm. Data are presented as mean ± SD. Source data are provided as a source data file.

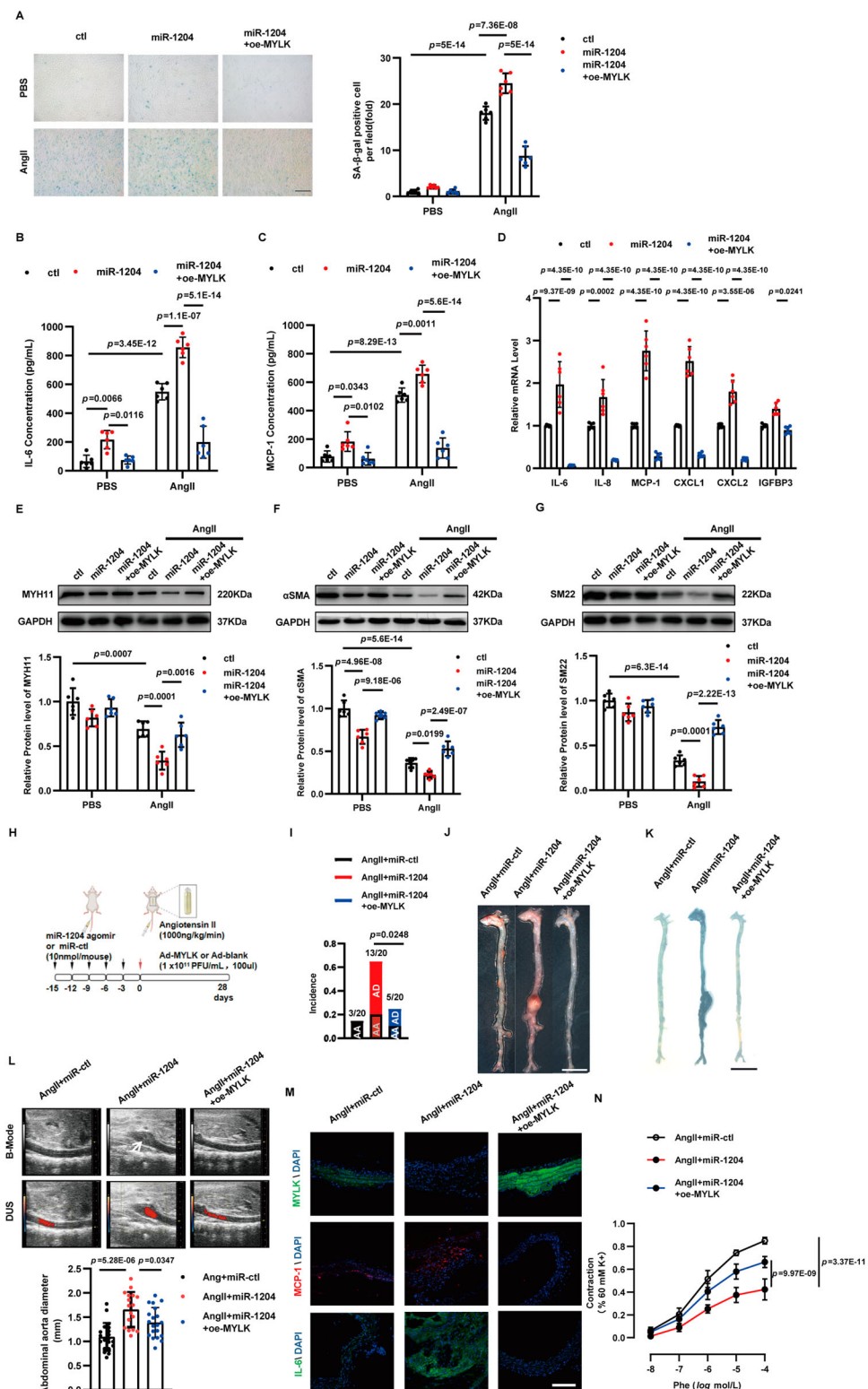

the somites and mesoderm, respectively) are also different, leading to structural and biochemical aortic heterogeneity. These differences make TAA more prone to rupture and dissection than AAA. In addition, many ADs do not form TAA before dissection, particularly in patients with multifaceted syndromes. Although these diseases are different, they share several pathogenic similarities, including proteolytic elastic tissue degeneration, smooth muscle dysfunction, and inflammation. Previous studies have shown that both sirtuin 1 and myocardin-related transcription factor A affect VSMC senescence and AAA formation[6,8].

Adenosine diphosphate induces VSMC senescence and promotes TAAD formation[10]. These findings suggest that aging is a common risk factor for AAD. However, the mechanism through which aging regulates AAD formation remains unclear. In the present study, we found that miR-1204 expression markedly increased in the plasma and aorta of older patients with AAD, including those with TAAD or AAA. We also found that miR-1204 expression was induced during cell senescence. Importantly, miR-1204 induces senescence in VSMC, forming a positive feedback loop that aggravates aging. Using both AngII and BAPN

**Fig. 6 | Myosin light chain kinase (MYLK) ameliorates miR-1204-induced senescence-associated secretory phenotype (SASP) and contractile phenotype loss in vitro and in vivo. A** Representative images of senescence-associated-β-galactosidase (SA-β-gal) staining (left) and quantification (right) in the indicated groups. $n = 6$ biological replicates. Statistical analysis was performed using a one-way ANOVA followed by Tukey's multiple comparison test. Scale bar represents 100 μm. **B** Interleukin 6 (IL-6) concentration in the culture supernatant of VSMCs in the indicated groups. $n = 6$ biological replicates. Statistical analysis was performed using a one-way ANOVA followed by Tukey's multiple comparison test. **C** Monocyte chemotactic protein 1 (MCP-1) concentration in the culture supernatant of VSMCs in the indicated groups. $n = 6$ biological replicates. Statistical analysis was performed using a one-way ANOVA followed by Tukey's multiple comparison test. **D** Quantification of the mRNA levels of SASP components in VSMCs transfected with miR-1204 mimics or miR-ctl in the indicated groups. $n = 6$ biological replicates. Statistical analysis was performed using a one-way ANOVA followed by Tukey's multiple comparison test. **E−G** Representative plots (upper panel) and quantification (lower panel) of immunoblot analysis of contractile markers, myosin heavy chain 11 (MYH11), α-smooth muscle actin (α-SMA), and smooth muscle protein 22 (SM22) in the indicated groups. $n = 6$ biological replicates. Statistical analysis was performed using a one-way ANOVA followed by Tukey's multiple comparison test.

**H** Scheme of AngII-induced aortic aneurysm and dissection (AAD) experiments. MiR-1204 agomir or miR-ctl (10 nmol) was administered by tail vein injection to 4-month-old male and female C57BL/6 mice every 3 d for a total of five times. Then the mice were administered MYLK adenovirus or blank adenovirus via tail vein prior to 4-weeks of AngII-infusion. **I** AAD incidence. AA aortic aneurysm, AD aortic dissection. Statistical analysis was performed using two-tailed Fisher's exact test. **J** Representative photographs of aortas in the indicated groups. Bar indicates 4 mm. **K** Representative images of SA-β-gal staining in the indicated groups. Scale bar indicates 4 mm. **L** Representative B-mode ultrasound and doppler ultrasound (DUS) detection of abdominal aortas (upper) and quantification of abdominal aorta diameters (below). $n = 17$ biological replicates for AngII + miR-1204 group. $n = 20$ biological replicates for other groups. Statistical analysis was performed using a one-way ANOVA followed by Tukey's multiple comparison test. The white arrow indicates an aneurysm. **M** Representative images of immunofluorescence staining of MYLK, MCP-1, and IL-6 in mouse aortas in the indicated groups. Scale bar represents 100 μm. **N** Contraction of isolated aortic rings from indicated groups in response to the phenylephrine (Phe) treatment. $n = 8$ biological replicates. Statistical analysis was performed using two-tailed repeated-measures ANOVA with Bonferroni's multiple comparison test. Data are presented as mean ± SD. Source data are provided as a source data file.

animal models, we demonstrated that miR-1204 aggravates AAD formation, and that aging induces AAD formation via miR-1204. This finding suggests that aging may have the same pathogenesis in all types of AAD formation and that miR-1204 may be the link between aging and AAD.

DNA damage is known to induce cell senescence. A previous study showed that DNA damage may be involved in aorta dilation[26]. Using a DNA damage model induced by AngII, $H_2O_2$, and daunorubicin, we found that miR-1204 expression is induced by DNA damage via the p53 pathway, suggesting that factors such as hypertension, oxidative stress, radiation, and chemical agents can induce miR-1204 expression. However, cellular damage does not directly lead to signs of aging. When the cumulative damage reaches a certain limit, cells acquire the SASP[27]. The SASP is composed of a series of cytokines, including pro-inflammatory cytokines, growth factors, chemokines, and matrix-remodeling enzymes. The SASP secretes various pro-inflammatory factors that affect cell and tissue biology[28]. Accumulating evidence suggests that the SASP is involved in various diseases, including tumors, diabetes, osteoarthritis, chronic obstructive pulmonary disease, kidney diseases, and chronic cardiovascular diseases[29,30]. In the present study, we found that miR-1204 increased the synthesis of SASP components in VSMCs, demonstrating that miR-1204 induces VSMC senescence to acquire SASP. This leads to cytokine/chemokine release, dedifferentiation, and vascular inflammatory infiltration, resulting in the aggravation of AAD. A previous study reported that miRNAs regulate cell senescence and inflammation, which supports our findings[31].

In the current study, we found that miR-1204 induces VSMC senescence to acquire SASP by inhibiting MYLK expression. Previous studies have demonstrated that heterozygous loss-of-function mutations in MYLK causes the dissection of the thoracic aorta[32,33]. This further supports our finding that miR-1204 exacerbates AAD by inhibiting MYLK, thereby inducing VSMCs to acquire SASP, and causing the loss of their contractile phenotype. Furthermore, we also found that miR-1204 reduced the expression of TGFβR2 in VSMCs, whereas MYLK increased the expression of TGFβR2 in VSMCs. These results are consistent with previous findings[22,34,35]. Taken together, our study shows that miR-1204 aggravates AAD formation by modulating MYLK, inducing VSMC to acquire SASP and lose their contractile phenotype.

In addition to MYLK, miR-1204 markedly suppressed the transcriptional expression of Ras homolog family member J (RHOJ) and 7-dehydrocholesterol reductase (DHCR7). RHOJ, a member of the RHO GTPase family, exhibits high endothelial cell-restricted expression in several different tissues[36]. Studies have demonstrated that RHOJ localizes to focal adhesions and regulates endothelial cell migration

and tube formation, while modulating actomyosin contractility and the number of focal adhesions[37]. To date, numerous studies have documented the crucial causal involvement of endothelial barrier function in the progression of AAD, encompassing disruptions in tight junctions and adherens junctions, as well as extracellular attachments like focal adhesions[38]. Despite the low expression levels of RHOJ in aortic VSMCs, future studies should focus on the regulation of RHOJ by miR-1204 in endothelial cells to further explore the effect of miR-1204 on endothelial barrier function and its role in AAD. DHCR7 catalyzes the conversion of 7-dehydrocholesterol to cholesterol, which is the final step in cholesterol synthesis and significantly affects cellular cholesterol homeostasis[39]. Loss of enzymatic activity results in accumulation of the substrate 7-dehydrocholesterol, which increases vitamin D production. The inhibition of dehydrocholesterol reductase has been shown to decrease cholesterol biosynthesis in VSMCs, leading to impaired adhesion, migration, and proliferation, thereby affecting their functional steady state[40]. Given that the effect of cholesterol biosynthesis on VSMCs characteristics is largely unknown, the influence of miR-1204 on VSMCs in this context warrants further investigation.

Our study also suggests that miR-1204 may be a potential biomarker for age-related AAD, as plasma miR-1204 levels are significantly increased in older patients with AAD. Currently, there are no preventive or predictive methods available for AAD. Although D-dimer levels are increased in patients with acute AD, they can not be used as markers to differentiate from pulmonary embolism cases, which also exhibits markedly elevated D-dimer levels[41]. Soluble suppression of tumorigenicity protein 2 (ST2) represents a promising diagnostic biomarker because it is significantly elevated in acute AD[42]. Similarly, it is non-specifically elevated in many inflammatory diseases, including fulminant myocarditis and heart failure[43,44]. More importantly, both soluble ST2 and D-dimer levels are normal prior to AD onset and increase only after the onset of AD, suggesting that they may not be appropriate for predicting AD occurrence. Soluble ST2 and D-dimer levels do not increase significantly in aortic aneurysms, including AAA and TAA. In general, the current research on biomarkers has neglected the role of aging and aging-related inflammation in the progression of AAD. This is particularly important because it can predict the development of AAD especially that of AD, in an aging populations. Although plasma succinate concentrations are elevated in AAA and AD, it remains unclear whether they are also elevated in TAA. In addition, this elevation is not specific to aging patients[45]. Importantly, plasma succinate levels are also increased in various diseases, including hypertension, ischemic heart disease, diabetes, obesity, inflammatory

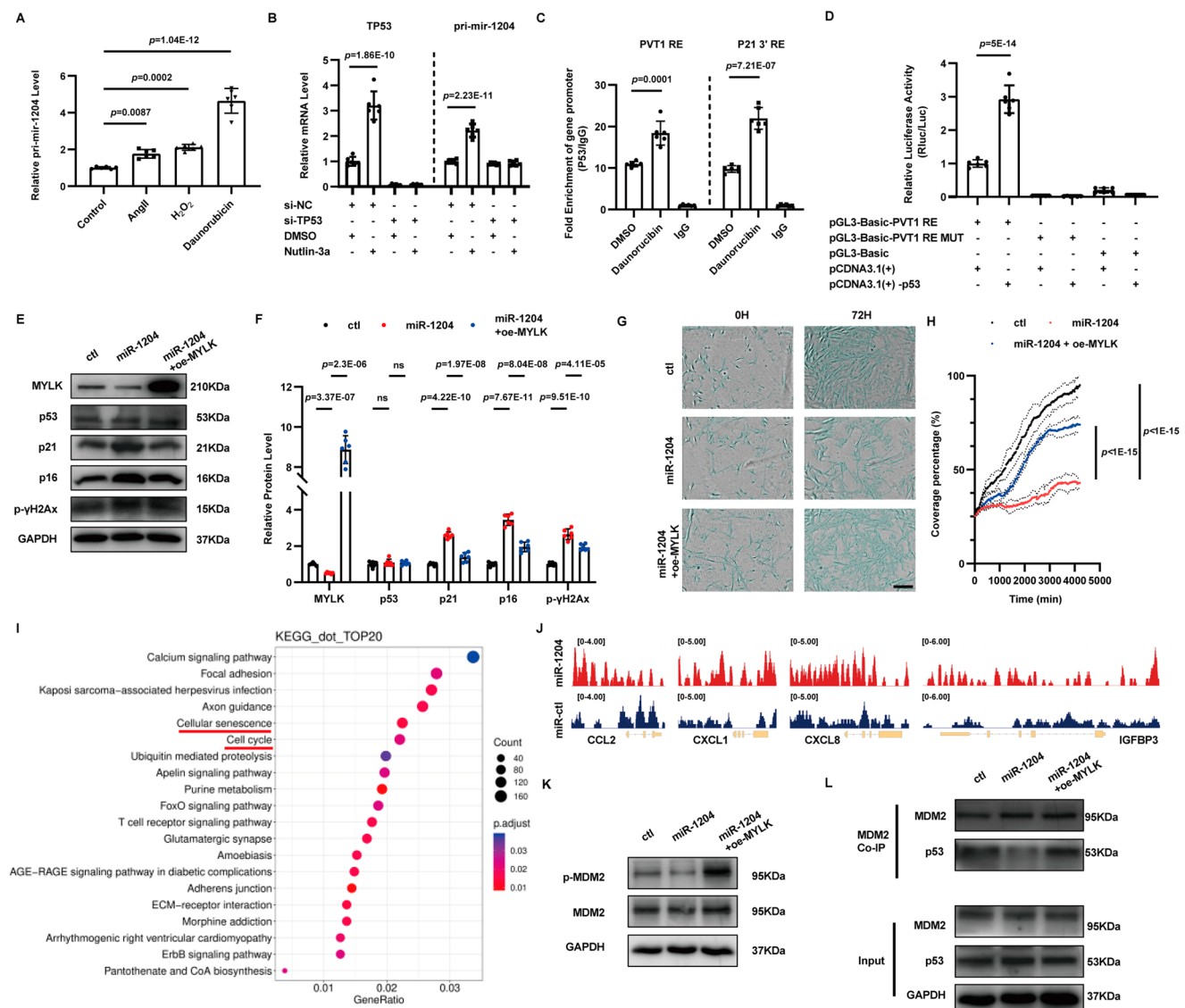

**Fig. 7 | P53 up-regulates miR-1204 through interaction with plasmacytoma variant translocation 1 (PVT1) response element in senescent vascular smooth muscle cells (VSMCs). A** Quantification of pri-mir-1204 levels in VSMCs treated with angiotensin II (AngII, $1 \times 10^{-6}$ M), hydrogen peroxide ($H_2O_2$, 30 μM) or daunorubicin (0.22 μM). $n = 6$ biological replicates. Statistical analysis was performed using a one-way ANOVA followed by Tukey's multiple comparison test. **B** Quantification of pri-mir-1204 levels in VSMCs treated with nutlin-3a with or without silencing TP53. $n = 6$ biological replicates. Statistical analysis was performed using a one-way ANOVA followed by Tukey's multiple comparison test. **C** Chromatin immunoprecipitation analysis quantifying DNA fragments co-immunoprecipitated with p53 at the PVT1 response element (PVT1 RE). The P21 3′ response element (P21 3′ RE) was used as a positive control. $n = 6$ biological replicates. Statistical analysis was performed using a one-way ANOVA followed by Tukey's multiple comparison test. **D** Luciferase activity confirms the interaction between p53 and PVT1 RE. $n = 6$ biological replicates. Statistical analysis was performed using a one-way ANOVA followed by Tukey's multiple comparison test. **E** Representative plots of immunoblot analysis of cellular senescence markers, p53, p21, p16, and p-γH2AX in control VSMCs, VSMCs transfected with miR-1204 mimics alone and VSMCs transfected with miR-1204 mimics plus MYLK adenovirus.

**F** Quantification of cellular senescence markers in the indicated groups by immunoblot analysis. $n = 6$ biological replicates. Statistical analyses were performed using a one-way ANOVA followed by Tukey's multiple comparison test expect for MYLK quantification. Statistical analysis was performed using two-tailed Welch's ANOVA, followed by Dunn's multiple comparisons for MYLK quantification. **G** VSMC division was monitored for 72 h in control VSMCs, VSMCs transfected with miR-1204 mimic alone, and VSMCs transfected with miR-1204 mimic and MYLK adenovirus. Scale bar represents 200 μm. **H** Quantification of VSMC division in the indicated groups. $n = 3$ biological replicates. Statistical analysis was performed using two-tailed repeated-measures ANOVA with Bonferroni's multiple comparison test. **I** KEGG analysis of target genes binding to p53 in VSMCs overexpressing miR-1204. Statistical analyses were performed using hypergeometric tests. **J** Distribution of reads from ChIP-Seq of SASP genes in miR-1204 overexpressing or control VSMCs. **K** Phosphorylation of MDM2 in the VSMCs from indicated groups. The experiment was independently repeated three times with similar results. **L** Expression of MDM2 and p53 in proteins pulled-down by MDM2-Co-IP and input immunoprecipitation in VSMCs from indicated groups. The experiment was independently repeated three times with similar results. Data are presented as mean ± SD. ns not significant. Source data are provided as a source data file.

diseases, and exercise, which are not easily differentiated from AAD[46,47]. Similarly, a previous study that screened circulating miRNAs as biomarkers to predict aortic aneurysm and aneurysm growth did not adequately analyze age-related effects[48]. More importantly, current literature suggests that miR-1204 expression is not increased in other disease states except in breast cancer and ovarian squamous cell carcinoma, which can be easily differentiated from AAD[49,50].

Limitations: Although our study suggests that miR-1204 is a potential biomarker for age-related AAD, the data were obtained from two centers. Whether miR-1204 is a potential biomarker of age-related

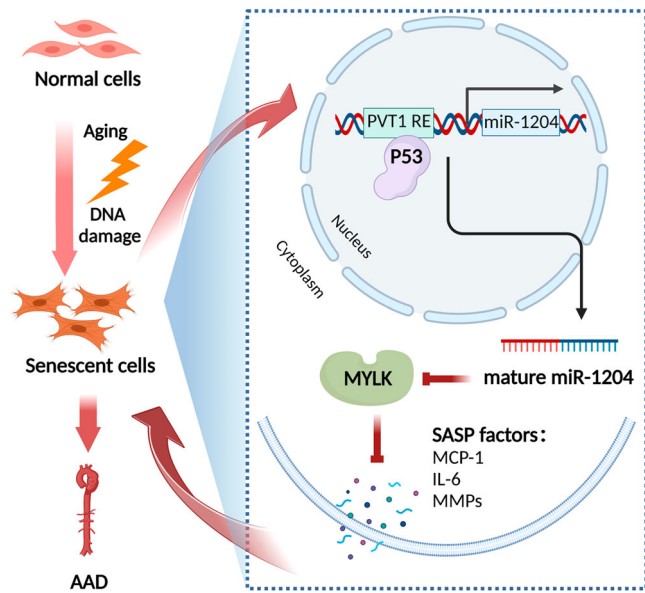

**Fig. 8 | Hypothetical working model.** Cell senescence induces the expression of miR-1204 through interaction of p53 with the plasmacytoma variant translocation 1 response element (PVT1 RE). MiR-1204 directly targets myosin light chain kinase (MYLK), leading to the acquisition of a senescence-associated secretory phenotype (SASP) by vascular smooth muscle cells (VSMCs) and loss of their contractile phenotype. MiR-1204 induces VSMC senescence by forming a positive feedback loop. This figure was created with Bio-Render.com.

AAD needs to be further studied in an independent validation cohort with a larger sample size from multiple centers.

In summary, the present study provides direct evidence that aging induces miR-1204 to inhibit MYLK, promoting VSMCs to acquire SASP and lose their contractile phenotype. This results in cytokine/chemokine release, vascular inflammatory infiltration, and VSMC dedifferentiation, leading to the aggravation of AAD formation. Additionally, miR-1204 promotes VSMC senescence to form a positive feedback loop that aggravates AAD formation (Fig. 8). Our findings revealed the molecular mechanism underlying aging-induced AAD. The p53/miR-1204/MYLK signaling axis represents a potential therapeutic target for AAD, and miR-1204 may be a biomarker for age-related AAD.

## Methods

### Ethics statement
This study was approved by the Ethics Review Board of the First Affiliated Hospital, Sun Yat-sen University, and Second Xiangya Hospital of Central South University. Written informed consent was obtained from all the human participants enrolled into this study. All animal experiments were approved by the Ethics Review Board and Animal Research Committee of the First Affiliated Hospital, Sun Yat-sen University.

### Study populations
In total, 272 patients (116 young patients and 156 elder patients) diagnosed with AAD and 158 healthy subjects (70 young, healthy subjects, 88 elder healthy subjects) were recruited for plasma miRNA detection. Patients were diagnosed with AAD on the basis of computed tomographic angiography. Patients with infectious diseases, tumors, renal failure and those had undergone surgery or severe trauma within the preceding three months were excluded. Healthy subjects who underwent physical examinations during the same period were enrolled. The participants fasted overnight prior to blood collection, and peripheral venous blood was acquired using ethylene diamine tetraacetic acid (EDTA) anticoagulant tubes. Following centrifugation

for 15 min at $1500 \times g$ at 4 °C, plasma was collected and stored at −80 °C until further use. Human AAD samples were obtained from patients with type A dissection who underwent open surgery. Normal aortic tissue was obtained from transplant donors. Aortic tissue specimens were separated and placed on ice. One portion was snap-frozen in liquid nitrogen for miRNA extraction. The other portion was fixed in 4% paraformaldehyde (PFA) and embedded in paraffin for sectioning and staining.

### MiRNA microarray
Microarray analysis was performed as described previously[51]. Briefly, plasma samples from young normal (<50 years old), elder normal (>50 years old), young patient (<50 years old), and elder patient (>50 years old) groups ($n = 3$ per group) were used to detect the miRNA profiles using miRCURY LNA Array (version 19.0) (Exiqon miRCURY LNA™ microRNA Array, 7th gen-hsa, mmu and rno). After RNA extraction, the samples were sent to KangChen Bio-tech (Shanghai, China) for microarray hybridization, data generation, and normalization according to the manufacture's guideline. Finally, the differentially expressed miRNAs were analyzed base on the threshold level (fold change ≥ 2, $P ≤ 0.05$) between the indicated groups, and hierarchical clustering of miRNAs was performed.

1. RNA extraction: Plasma was separated from blood collected with the anticoagulant EDTA. Total RNA was isolated using TRIzol™ (Invitrogen) and miRNeasy™ Mini Kit (Cat.74104, Qiagen, USA), which efficiently recovered all RNA species, including miRNAs. The RNA quality and quantity were measured using an ND-1000 NanoDrop spectrophotometer.
2. RNA labeling and array hybridization: After quality control tests, the miRNAs were labeled using the miRCURY™ Hy3™/Hy5™ Power Labeling Kit (Exiqon) according to the manufacturer's instructions. In brief, purified RNA was mixed with CIP buffer and CIP (Exiqon) for 30 min at 37 °C. The reaction was terminated by incubation for 5 min at 95 °C. Then labeling buffer, fluorescent label (Hy3™), DMSO and labeling enzyme were added to the mixture. The labeling reaction was incubated for 1 h at 16 °C. After stopping the labeling reaction, the Hy3™-labeled samples were hybridized on the miRCURY™ LNA Array (version 19.0) according to manufacturer's instructions. Slides were scanned using an Axon GenePix 4000B microarray scanner (Axon Instruments, CA, USA).
3. Data analysis: Scanned images were imported into GenePix Pro 6.0 software (Axon) for grid alignment and data extraction. Replicated miRNAs were averaged and miRNAs with intensities ≥ 30 in all samples were chosen to calculate the normalization factor. The expression data were normalized by the median normalization. After normalization, significantly differentially expressed miRNAs between indicated groups were identified based on their threshold levels(fold change ≥ 2, $P ≤ 0.05$). Finally, hierarchical clustering was performed to identify distinguishable miRNA expression profiles among the samples.

### miRNA quantitative real-time PCR (qRT-PCR)
Plasma miRNA levels in all 430 participants were verified as previously reported[52]. Briefly, plasma miRNA was extracted using a miRNeasy Serum/Plasma Kit (Cat.217184, Qiagen, USA), and RNA purity was examined using a NanoDrop 2000 spectrophotometer by measuring the ratio of absorbance at 260 and 280 nm. Cel-miR-39 was used as a spike-in control. Reverse transcription and qRT-PCR were performed with Bulge-Loop™ miRNA qRT-PCR Starter Kit (C10211-2, Ribobio, China) using a Bio-Rad CFX96 system. Relative quantification was performed using the ΔΔCt method, in which $\Delta Ct = Ct_{targeted\ miRNA} - Ct_{miR-39}$. Data are presented as the difference in value relative to that of the young normal group.

For tissue miRNA extraction, a Tissue RNA Purification Kit Plus (Cat.RN002plus, EScience, China) was used according to the

manufacturer's instruction. Reverse transcription and qRT-PCR were performed with Bulge-Loop™ miRNA qRT-PCR Starter Kit (C10211-2, Ribobio, China) using a Bio-Rad CFX96 system. The data were normalized to U6 as an endogenous control and presented as the fold change relative to that of the young normal group. Bulge-loop miRNA primers were designed and purchased from RiboBio Corporation (Guangzhou, China), and their sequence were proprietary. The primers used in this study are listed in Supplemental Table 6.

## ISH

miR-1204 was detected in formalin-fixed paraffin-embedded (FFPE) samples by ISH using miRCURY LNA™ miRNA Detection Probe (Cat.339111, Qiagen, USA) according to the manufacturer's instructions. Arterial samples were fixed in 4% PFA for at least 24 h and embedded in paraffin. Serial sections (3–5 μm thick) were made. Sections were deparaffinized and hydrated by exposure to xylene and graded alcohol followed by water. Then the samples were treated with proteinase K for 10 min at 37 °C. The samples were incubated with hybridization solution for 2–4 h at 42 °C before hybridization. Next, a hybridization mix containing 20 nM double-DIG LNA miR-1204 probe was added. Hybridization was carried out at 42 °C overnight, followed by stringent washes using graded SSC. The slides were incubated with blocking buffer for 30 min at 37 °C and then with anti-DIG reagent at 4 °C overnight. Finally, NBT/BCIP substrate was used to the sections for color development. Samples were imaged using a photomicroscope (Nikon DS-U3, Japan). The sequence of the miR-1204 Detection Probe is proprietary. The primers sequences used in the study are listed in Supplemental Table 6.

## Immunofluorescence staining

The paraffin sections of human aortic tissues, the samples were dewaxed and rehydrated. Sections were permeabilized in freshly prepared 0.5% TritonX-100 and blocked with 5% normal goat serum (NGS). Then, the tissues were incubated with the following primary antibodies: MYH11 (1/100, Cat.ab53219, Abcam, UK), α-SMA (1/200, Cat.19245, Cell Signaling Technology, USA), SM22 (1/100, Cat.ab14106, Abcam, UK), MYLK (1/50, Cat.sc-365352, Santa Cruz Biotechnology, USA), MCP-1 (1/50, Cat.ab214819, Abcam, UK), and CXCL1 (1/100, Cat.12335-1-AP, Proteintech, USA) overnight at 4 °C, followed by incubation with the corresponding secondary antibody: anti-mouse IgG H&L (Alexa Fluor® 488)(1/1000, Cat.4408, Cell Signaling Technology, USA) or Anti-rabbit IgG H&L (Alexa Fluor® 555)(1/1000, Cat.4413, Cell Signaling Technology, USA) for 1 h. Nuclei were counterstained with DAPI. Fluorescence signals were captured using a laser scanning confocal microscope (Zeiss LSM 780, Germany).

The mouse aorta was excised and embedded in OCT compound (Cat.4583, Sakura, Japan) and serial cryosections (8 μm thick) were prepared. The frozen sections were fixed in 4% PFA permeabilized with freshly prepared 0.5% TritonX-100 and blocked with 5% NGS. Then, the tissues were incubated with the following primary antibodies: CD68 (1/100, Cat.ab283667, Abcam, UK), MYH11 (1/100, Cat.ab53219, Abcam, UK), α-SMA (1/200, Cat.19245, Cell Signaling Technology, USA), SM22 (1/100, Cat.ab14106, Abcam, UK), IL-6 (1/50, Cat.ab233706, Abcam, UK), MCP-1 (1/50, Cat.ab214819, Abcam, UK) and MYLK (1/50, Cat.sc-365352, Santa Cruz Biotechnology, USA) overnight at 4 °C, followed by incubation with the corresponding secondary antibody: anti-mouse IgG H&L (Alexa Fluor® 488)(1/1000, Cat.4408, Cell Signaling Technology, USA) or Anti-rabbit IgG H&L (Alexa Fluor® 555)(1/1000, Cat.4413, Cell Signaling Technology, USA) for 1 h. Nuclei were counterstained with DAPI. Fluorescence signals were captured using a laser scanning confocal microscope (Zeiss LSM 780, Germany).

## Animals

Male and female C57BL/6 mice were used in the experiments. Mice were obtained from the Laboratory Animal Center of Sun Yat-sen University (Guangzhou, China). Mice were housed in cages with 50% humidity at 20 °C temperature with a 12 h light/dark cycle, and provided the standard rodent chow diet and water. Special attention was paid to animal welfare and the minimization of suffering. To achieve 90% power to detect a difference between the groups with a significance level of 0.05, the PASS software 15 (NCSS LLC., Kaysville, UT, USA), was used to estimate the animal sample size based on AAD incidence in the pre-experiment. All animal experiments were approved by the Ethics Review Board and Animal Research Committee of the First Affiliated Hospital, Sun Yat-sen University.

## Disease models

1. AngII-induced AAD model: An osmotic pump (Model 2004, Alzet, USA) containing either AngII (1000 ng/kg/min, Cat.A9525, Sigma Aldrich, USA) or saline was introduced into 4-month-old C57BL/6 mice, as previously described[20]. A miR-1204 agomir (Cat.miR40005868, Ribobio, China) or miRNA control (10 ng) was delivered by tail vein injection every three days for a total of five times prior to AngII-infusion to assess their effects on AAD development. Cy5-labeled miR-1204 agomir (Cat.miR4200521013014, Ribobio, China) was delivered by tail vein injection, and fluorescent images were obtained after 6 h using an IVIS imaging system (Caliper Life Sciences, MA, USA) at excitation and emission wavelengths of 649 and 680 nm, respectively. The miR-1204 agomir sequence is listed in the Supplemental Table 5. The miRNA control sequence was proprietary, and is listed in Supplemental Table 6.

2. BAPN-induced AAD model: Three-week-old C57BL/6 mice were fed a normal diet and administered freshly prepared BAPN (1 g/kg/d, Cat.A3134, Sigma-Aldrich, USA) solution dissolved in the drinking water for four weeks as described previously[53]. A single injection of LNA anti-miR-1204 (Cat.339204, Qiagen, USA) or scrambled miRNA was administered via the tail vein one day prior to AAD induction[14]. The sequences of custom-synthesized LNA anti-miR-1204 and scrambled miRNA are proprietary and are listed in the Supplemental Table 6.

Vital parameters of the mice, including weight and blood pressure were monitored on days 0, 7, 14, 21, and 28. SBP was measured in conscious mice using the pulse-based tail-cuff method. The tail was constrained using a cuff. A BP-2010A intravascular blood pressure transducer (Softron, Beijing, China) was used to indirectly record the blood pressure. SBP was measured at least eight times for each mouse.

The maximal diameter and flow distribution of the ascending aorta, aortic arch, descending aorta, and abdominal aorta were measured in vivo on day 28 using a Vevo 2100 ultrasound system (Visual Sonics, Toronto, Canada). Mice were euthanized using 1.5% isoflurane and placed on a heated platform in the supine position. The mouse limbs were kept close to the electrode and the electrocardiogram (heart rate 220-420 bpm) was recorded simultaneously. Imaging was performed using a MS400 transducer with a frequency of 30 MHz. Right parasternal approaches were selected as the standard ultrasound modes for mice. The transducer was placed on the right edge of the sternum at a 45-degree angle relative to the chest. Measurements of aortic diameters were recorded during end-diastole. Multiple cardiac cycles (at least six) were recorded, and the acquired images were stored for further analysis. All measurements were collected by one observer to limit bias, whereas another independent observer analyzed the records. AA was defined as a 50% increase in the aortic diameter compared to the mean diameter of the control group. The observation of intimal injury to form true and false lumens separated by an intimal flap was defined as AD. The presence of aortic rupture and premature death, and hematoma within the aortic wall detected during gross examination were also considered as dissection.

## Morphology

The mice were euthanized using an overdose of isoflurane, perfused with PBS, and the aortas were excised. They were fixed with 4% PFA solution before being embedded in paraffin for sectioning. Tissues were sectioned (8 μm thick) and stained with hematoxylin & eosin (H&E) or EVG staining. H&E staining was used to observe the overall morphology and EVG was used to discern the breakdown of the elastic layers. Degradation of the elastic layers was scored based on the extent of elastin degradation-grading as follows: grade 1, no degradation; grade 2, mild elastin degradation; grade 3, severe elastin degradation; grade 4, aortic rupture. All the images were captured using a scanner (KF-PRO-020, KFBIO).

## SA-β-gal staining

SA-β-gal activity was determined using a SA-β-gal Staining Kit (Cat.9860, Cell Signaling Technology, USA) as previously reported[26]. Briefly, anesthetized mice were subjected to antegrade perfusion with PBS followed SA-β-gal staining solution ((titrated to pH=6.0) via the left ventricle. Aortas were then excised, incubated in SA-ß-gal staining solution at 37 °C overnight. Tissue was then frozen, embedded in OCT, and cryosections (8 μm thick) were imaged microscopically (Leica, Germany).

VSMCs were plated in 6-well plates. After different treatments, the medium was removed and the cells were washed with PBS three times. The cells were then fixed in a fixative solution for 10 min at room temperature. After washing twice with PBS, the cells were incubated with staining solution (titrated to pH=6.0) and incubated at 37 °C overnight. The cells were imaged with a Leica microscope and the number of positively stained (blue) cells were assessed[54].

## Aortic ring myography

Myography was performed as previously described[55]. Briefly, the aortic rings were dissected and isolated from the mice following euthanasia. The aortic rings (2 mm wide) were transferred to Krebs-solution (pH 7.4,119 mM NaCl; 25 mM NaHCO$_3$; 1.6 mM CaCl$_2$; 4.7 mM KCl; 1.2 mM KH$_2$PO$_4$; 1.2 mM MgSO$_4$·7H$_2$O; and 11.1 mM D-glucose) and equilibrated for 1 h. The solution was changed every 15 min. Subsequently, the aortic rings were subjected to vascular tension experiments. The cumulative dose-response to phenylephrine (Cat. HY-B0769, Med-ChemExpress, USA) was used to characterize vasocontraction. The data are expressed as percentages of KCl-induced contraction.

## Northern blotting

Aortas were dissected from PBS-perfused animals and total RNA was isolated using the TRIzol™reagent (Cat. 15596018, Invitrogen, USA). For Northern blot analysis, 2 μg of total RNA was separated on a 6% polyacrylamide gel and blotted onto a Hybond™ XL hybridization membrane. Hybridization was performed in hybridization buffer at 30 °C overnight. Blots were washed two times in 2 x SSC/0.1% SDS at 25 °C for 5 min and then at 42 °C for 15 min. Signals were detected using a Kodak Gel Logic 1500 Imaging System.

## Cell culture

Primary human aortic VSMCs (HASMCs) were purchased from Scien-Cell (Cat. No. 6110, Carlsbad, CA, USA), and cultured in smooth muscle cell medium (SMCM, Cat.No.1101, ScienCell) supplemented with 2% fetal bovine serum (FBS), 1% penicillin/streptomycin, and growth factor. According to the manufacturer, The HASMCs were sourced from an unspecified region of the aorta, and could contain cells from any region of the aorta. HEK-293T cells were purchased from Procell (Cat.CL-0005, China) and cultured in Dulbecco's modified Eagle's medium (DMEM; Cat.C11995500BT, Gibco, USA) supplemented with 10% FBS and 1% penicillin/streptomycin. All cultures were maintained in a humidified 5% CO$_2$ atmosphere at 37 °C. Prior to treatments, the cells (passages 4–6) were washed with DPBS, trypsinized, counted, and plated in 6-, 12-, 24-, 48-, 96-well plates depending on the experiments[56,57]. All cells were serum-starved overnight to induce cell synchronization prior to treatment.

## Cell transfection and transduction

Small interfering RNAs (siRNAs) targeting MYLK and TP53, and miR-1204 mimics were designed and synthesized by RiboBio Corporation (Guangzhou, China). A mixture of the three siRNAs (50 nM) or miRNA (50 nM) was transfected into cells using Lipofectamine™ RNAiMAX Reagent (Cat.13778030, Invitrogen, USA) according to the manufacturer's protocol. Non-targeting siRNAs or miRNAs were used as controls. After 24 h of transfection, the cells were treated with PBS or AngII ($1 \times 10^{-6}$ mol/L) for 24 h for some experiments as described below. The siRNA and miRNA sequences are listed in the Supplemental Table 5.

MYLK overexpressing adenoviruses were designed and purchased from Hanbio Biotechnology (Shanghai, China). Once the cells reached 30–50% confluence in 6-well plates, they were infected with adenoviral particles at a multiplicity of infection (MOI) of 10. The medium containing the adenovirus was changed after 6 h of transduction. The cells were cultured for at least 24 h prior to further experiments.

## TMT proteomics

Proteomic analysis was performed as described previously[58]. Briefly, VSMCs transfected with miR-1204 mimic or miR-control were harvested ($n = 3$/group). Samples were then sent to Jingjie PTM BioLabs (Zhejiang, China) for protein extraction, trypsin digestion, TMT labeling, high-pressure liquid chromatography (HPLC) fractionation, LC-MS/MS, and data analysis according to standard protocols. Finally, the differentially expressed proteins were filtered based on present threshold levels (fold change ≥ 1.3, $P < 0.05$) between two groups and hierarchical clustering was performed. Protein-protein interaction-associated functions were identified based on GO and KEGG pathway analyses.

1. Protein preparation: VSMCs transfected with miR-1204 mimic or miR-control were harvested using cell scrapers. The cells were resuspended in 8 M urea supplemented with 1% proteinase inhibitor and sonicated thrice on ice. After centrifugation for 10 min at 12,000 x g at 4 °C, the supernatant was collected and protein concentration was measured using a bicinchoninic acid (BCA) Kit (Cat.23225, Thermo Fisher Scientific, USA) according to the manufacturer's instructions.

2. Tryptic digestion: For digestion, reductive alkylation of the protein was carried out using 5 mM dithiothreitol and 11 mM iodoacetamide. The protein sample was then diluted and trypsin was added at trypsin-to-protein mass ratio of 1:50 and digested overnight at 37 °C.

3. TMT labeling: Trypsin-digested peptides were desalted using a Strata X C18 SPE column (Phenomenex) and dried in a vacuum container. The dried peptide was reconstituted in 0.5 M TEAB and mixed with TMT reagent for 2 h at room temperature, desalted, and vacuum-dried.

4. HPLC fractionation: The tryptic peptides were fractionated by high pH reverse-phase HPLC using a C18 column with 5 μm particle size, 4.6 mm ID, and 250 mm length. The peptides were then dried in a vacuum container.

5. LC-MS/MS analysis: The tryptic peptides were dissolved in 0.1% formic acid and subjected to nano-spray ionization (NSI), followed byMS/MS in a Q ExactiveTM HF-X (Thermo, USA) coupled online to UPLC. An electrospray voltage of 2.1 kV and an m/z scan range of 350–1600 was used for the full system scan. Intact peptides were detected in Orbitrap at a resolution of 120,000 and fragments were detected at a resolution of 30000. The intact peptides or fragments were selected for MS/MS analysis using an NCE setting of 28. The data-dependent procedure alternated

between one MS scan followed by 20 MS/MS scans with 30.0 s dynamic exclusion. The automatic gain control (AGC) was set to 1E5. The first fixed mass was set to 100 m/z. The resulting MS/MS data were processed using the MaxQuant search engine (v.1.6.15.0).

6. Data analysis: To further understand the functions and features of the identified and quantified proteins, we annotated the functions or features of proteins from several different categories, including GO, KEGG pathway, and COG/KOG functional classification and subcellular localization. All differentially expressed proteins were searched against the STRING database version 11.0 for protein-protein interactions. We fetched all interactions that had a confidence score > 0.7 (high confidence).

## Total RNA extraction, cDNA synthesis, and qRT-PCR
VSMCs were plated in 6-well plates. Following the different treatments, total RNAs were extracted using the RNA-Quick Purification Kit (Cat.RN001, EScience, China) according to the manufacturer's instructions. cDNA was synthesized with the NovoScript® Plus All-in-one 1st Strand cDNA Synthesis SuperMix (Cat.E047, Novoprotein, China). qRT-PCR was performed using a Bio-Rad CFX96 system. Relative gene expression was calculated and normalized against that of GAPDH. Data are presented as fold-change relative to that in the control groups. The primer sequences are listed in the Supplemental Table 5.

## Western blotting
VSMCs were cultured in 6-well plates as described above. Following the different treatments, cellular proteins were harvested for western blot analysis as previously reported[59,60]. Briefly, cells were scraped with 1×radioimmunoprecipitation assay (RIPA) buffer (Cat.9806, Cell Signaling Technology, USA) supplemented with 1% proteinase inhibitor. After centrifugation at 12,000 x g for 10 min at 4 °C, the supernatant was collected and the protein content was measured using BCA Kit (Cat.23225, Thermo Fisher Scientific, USA). Equal amounts of protein were mixed with the loading buffer and boiled for 10 min. The samples and molecular weight markers were loaded on gels and separated by sodium dodecyl sulfate-polyacrylamide gel electrophoresis (SDS-PAGE). The proteins were transferred from the gel to a polyvinylidene difluoride (PVDF) membrane. Then, the membranes were incubated with the primary antibody against: TGFβR2 (1/1000, Cat.79424, Cell Signaling Technology, USA), SMAD2/3 (1/1000, Cat.8685, Cell Signaling Technology, USA), MYH11 (1/1000, Cat.ab53219, Abcam, UK), α-SMA (1/1000, Cat.19245, Cell Signaling Technology, USA), SM22 (1/1000, Cat.ab14106, Abcam, UK), MYLK (1/100, Cat.sc-365352, Santa Cruz Biotechnology, USA), p53 (1/1000, Cat.2524, Cell Signaling Technology, USA), p16 (1/1000, Cat.92803, Cell Signaling Technology, USA), and GAPDH (1/1000, Cat.60004-1-Ig, Proteintech, USA) overnight at 4 °C, followed by incubation with horseradish peroxidase-linked secondary antibody: anti-mouse IgG HRP-linked antibody (1/10000, Cat.7076, Cell Signaling Technology, USA) or anti-rabbit IgG HRP-linked antibody (1/10000, Cat.7074, Cell Signaling Technology, USA) for 1 h. HRP substrate was added for chemiluminescent detection and images of the protein bands were captured with an Amersham Imager 600 (GE Healthcare). The densities of the bands were quantified using Adobe Photoshop 2020 software. Data are presented as fold-change relative to that in the control groups.

## Co-immunoprecipitation
VSMCs were cultured in T75 flasks. Following the different treatments, the cells were scraped into 1x Cell Lysis Buffer (Cat. 9803S, Cell Signaling Technology, USA) supplemented with 1% proteinase inhibitors. After centrifugation for 10 min at 12,000 × g at 4 °C, the supernatant was collected and protein concentration was measured using a BCA Kit (Cat.23225, Thermo Fisher Scientific, USA). Protein A/G magnetic beads (Cat. No. B23201, Bimake, USA) were incubated with anti-MDM2 (Cat. ab226939, Abcam, UK) or IgG (Cat. 2729, Cell Signaling Technology) antibodies for 30 min at room temperature. Next, the protein supernatant was incubated with antibody-magnetic beads complexes at 4 °C overnight. Following several washes, the SDS-PAGE loading buffer was added to the immunocomplexes, and the immunocomplexes were heated at 95 °C for 5 min. Finally, the denatured proteins were detected by western blot analysis, as described above.

## 5-Ethynyl-2′-deoxyuridine proliferation assay
EdU proliferation assays were performed using a Cell-Light EdU Apollo567®In Vitro Kit (Cat. C10310-1, RiboBio, China) according to the manufacturer's instructions. VSMCs were cultured in 48-well plates. Following different treatments, VSMCs were incubated with 50 µM EdU solution for 2 h. Then, the cells were fixed in 4% PFA and permeabilized in freshly prepared 0.5% Triton X-100. After washes with PBS, the cells were incubated with Apollo®567 reagent for 30 min and then washed with PBS containing with 0.5% Triton X-100. Finally, the cells were counterstained with Hoechst33342 and imaged using a fluorescent microscope (Leica, Germany).

## Collagen gel contraction assay
VSMC contractility was measured using a Cell Contraction Assay Kit (Cat. CBA-021, USA, Cell Biolabs) according to the manufacturer's instructions. The cells were harvested and resuspended in culture medium at a density of $5 \times 10^5$ cells/mL. A collagen lattice was prepared by mixing the cell suspension and ice-cold collagen gel solution in a volume ratio of 1:4. Then, 0.5 mL of the cell-collagen mixture was added to a 24-well plate and incubated for 1 h at 37 °C. After collagen polymerization, 1.0 mL of culture medium was layered on top of each collagen gel lattice. After 24 h, the plates were scanned. For data analysis, the gel area in each well was analyzed using Image J software (NIH, Bethesda, MD, USA).

## Wound healing assay
Wound healing migration assays were used to assess VSMCs migration. Briefly, VSMCs growing in monolayer in a 12-well plate were wounded by manually scraping the cells with a 200 µL pipette tip. The medium was replaced with serum free medium. At 0 and 18 h, migratory cells were observed using a Lionheart FX automated microscope (BioTek, Vermont, USA). The average scraped area of each well under each condition was measured and analyzed using the Image J software (NIH, USA).

## Transwell migration assay
Macrophage Transwell migration assays were performed using 8.0 µm Transwells (Cat. 3422, Corning, USA). $1 \times 10^5$ THP1 cells were added to the upper compartment of the chamber and the cells were treated with phorbol 12-myristate 13-acetate (PMA, 100 ng/mL) for 48 h to induce cell differentiation. After incubation for 24 h, the insert chamber was transferred to another well containing the conditioned medium in the lower chamber as a chemoattractant. After 12 h, the cells on the upper surface were removed with swabs, and the cells on the underside (invaded cells) were fixed and stained with 0.1% crystal violet. The mean number of cells on the lower side of the membrane was counted using a light microscope (six independent experiments were performed).

## Flow cytometry assay
Macrophages were analyzed using anti-CD80 PE/Cyanine7 antibody (Cat.305218, Biolegend, USA). All flow cytometry experiments were performed with a Beckman Coulter CytoFLEX platform using the CytExpert software. The optimal settings for each experiment were determined based on unstained samples. Live cells were gated based on side and forward scatter of the cells. Doublets were excluded based

on SSC-H vs. SSC-W plots. FlowJo software (version 10.8.1; BD, USA) was used to analyze the data.

## Enzyme-linked immunosorbent assay

The concentrations of IL-6 and MCP-1 in the cell culture supernatant were measured using competitive ELISA Kits (Cat.ab178013 and Cat.ab179886, Abcam, UK). Briefly, diluted cell culture supernatant (50 μL) was co-incubated with 50 μL capture and detector antibody cocktail in a 96-well plate for 1 h at room temperature on a plate shaker. The plate was washed twice and incubated with 100 μL of TMB development solution for 10 min in the dark. The reaction was stopped with a stop solution and the absorbance was measured at 450 nm.

## Gelatin zymography

Gelatin zymography was performed as described previously[61]. VSMCs were cultured in 6-well plates. Following the different treatments, the conditioned media were collected and centrifuged to exclude dead cells. The protein concentration was measured and equal amounts of protein samples were then mixed with loading buffer and electrophoresed on SDS-PAGE gels containing 1 mg/mL gelatin. The gels were washed twice with washing buffer (2.5% TritonX-100, 50 mM pH 7.5 Tris-HCl, 5 mM $CaCl_2$, and 1 μM $ZnCl_2$), incubated for 24 h with incubation buffer (1% TritonX-100, 50 mM pH 7.5 Tris-HCl, 5 mM $CaCl_2$, and 1 μM $ZnCl_2$) at 37 °C and finally stained with Coomassie brilliant blue.

## Luciferase assay

For luciferase assays, pmiR-RB-Report™ vectors containing MYLK 3'UTR target sequence, mutated sequence or deleted sequence were designed and purchased from RiboBio Corporation (Guangzhou, China). HEK-293T cells ($1 \times 10^4$) were cultured in DMEM supplemented with 10% FBS in 96-well plates for 24 h and then co-transfected with plasmids and either 100 nmol/L of miR-1204 mimic or miRNA control using Lipofectamine™ 3000 (Cat.L3000015, Invitrogen, USA). After 48 h, the cells were lysed and luciferase activities were measured using the Dual-Glo™ Luciferase Assay Kit (Cat.E2920, Promega, USA). Firefly luciferase activity was normalized to the Renilla luciferase activity. The MYLK 3'UTR sequences are shown in Fig. 5E.

For the PVT1 response element luciferase assay, wild-type or mutated PVT1 response elements were amplified by PCR and cloned into the KpnI-HindIII sites of the luciferase reporter pGL3-Basic. These constructs were co-transfected along with empty vector or p53-containing plasmids into VSMCs using Lipofectamine™ 3000 (Cat.L3000015, Invitrogen, USA). The pRL-TK plasmid containing Renilla luciferase was co-transfected for normalization. After 48 h, the cells were lysed and the luciferase activities were measured using the Dual-Glo™ Luciferase Assay kit (Cat.E2920, Promega, USA). The sequences used were as follows: wild-type PVT1 response element: 5′ CGACAAGTTGAGACTTGTTC 3′; mutant PVT1 response element: 5′TAGTGGACCAGAGTCCACCT 3′.

## Chromatin immunoprecipitation assay

ChIP assays were performed as described previously[62]. Briefly, VSMCs were cultured in 150 mm dishes, treated for 8 h with 0.22 μM daunorubicin or DMSO and subjected to ChIP using SimpleChIP® Plus Enzymatic Chromatin IP Kit (Cat. 9005, Cell Signaling Technology, USA). Formaldehyde (37%) was added to each dish containing 20 mL medium to crosslink the proteins to DNA. The cells were then washed and scraped into ice-cold PBS containing a protease inhibitor cocktail, followed by nuclear preparation and chromatin digestion. DNA fragment size and concentration were determined by electrophoresis and Nanodrop spectrophotometry, respectively. Chromatin was immunoprecipitated using ChIP grade anti-p53 (Cat.ab1101, Abcam, UK) or IgG (Cat.2729, Cell Signaling Technology, USA) antibody conjugated to protein G magnetic beads. After purification using spin columns, the precipitated chromatin DNA was quantified using qRT-PCR to determine the binding of the PVT1 or p21 response element to p53 or IgG. The primer sequences are listed in the Supplemental Table 5.

## ChIP-seq

VSMCs transfected with miR-1204 mimic or miR-control were harvested. Samples were then sent to Guangzhou SaiCheng Bio Co. Ltd. (Guangzhou, China) for sample preparation, sequencing and computational analysis according to the standard protocol.

1. Sample preparation and sequencing: $5–6 \times 10^7$ cells were cross-linked with 1% formaldehyde for 10 min at room temperature. The reaction was stopped with 2.5 M glycine at room temperature for 7 min. Nuclear lysates were sonicated to yield 200-500 bp DNA fragments and co-incubated with anti-p53 antibody (Cat.ab1101, Abcam, UK) at 4 °C overnight. Protein A/G Dynabeads (10015D, Life Technologies, USA) were blocked with PBS containing 1% BSA at 4 °C. Blocked beads and nuclear lysates were then co-incubated for 4 h at 4 °C. Beads were washed six times with RIPA buffer. The precipitated DNAs was eluted, reverse-crosslinked, and purified. ChIP DNA was then converted into Illumina sequencing libraries using the KAPA Hyper Prep Kit library preparation protocol. Multiplex adapters and TruSeq indices were used. The 150 bp paired end sequencing was performed using an Illumina NovaSeq 6000 (Novogene, Beijing, China).

2. Read mapping and peak calling: Sequenced reads (FASTQ format) WERE mapped using Bowtie2 (Version 2.5.0). Peak-calling identified significantly enriched loci (peaks) in the genome. Peak-calling was performed with using MACS2 software (version 2.1.0).

3. Visualization: ChIPseeker (version 3.17) was used to visualize the coverage of the ChIP seq data, peak annotation, average profile and heatmap of peaks bound to the TSS region. The proximal TSS (0–3 kb) is related to specific gene transcriptional regulation functions, and the distribution of the peak in the proximal TSS was statistically analyzed. ClusterProfiler software was used to conduct KEGG pathway enrichment analysis for differential gene sets.

## Statistical analyses

Quantitative data obtained from this study are expressed as means ± standard deviation (SD). Normality and equal variance were assessed for quantitative data using the Shapiro−Wilk and F tests to select a parametric or nonparametric test. Differences among the test groups that passed the aforementioned tests were determined using Student's *t* test for two groups or one-way analysis of variance (ANOVA) followed by Tukey's multiple comparison test for more than two groups. For data that did not pass the tests for normality and equal variance, nonparametric tests were used. The Mann−Whitney U test was used for two groups or the Kruskal−Wallis test with Dunn's multiple comparison test for more than two groups. Those that passed the normality tests but not the equal variance tests were analyzed using a parametric test with Welch's correction. Fisher's exact test was used to compare the incidence of AAD, and the log-rank test was used for survival comparisons between the groups. Repeated-measures ANOVA with Bonferroni's multiple comparisons test was used for weekly SBP analysis. Statistical significance was set at $P$-value < 0.05. Statistical analyses were performed using GraphPad Prism (version 9.0) and SPSS (version 25, IBM Corp., Armonk, NY, USA).

## Reporting summary

Further information on research design is available in the Nature Portfolio Reporting Summary linked to this article.

## Data availability

The data that support the findings of this study are available within the article and its supplementary information/source data or from the

corresponding authors upon request. Mass spectrometry proteomics data were deposited to the ProteomeXchange Consortium via the PRIDE partner repository with the dataset identifier PXD048658. Microarray data were deposited in Gene Expression Omnibus (GEO) using the dataset identifier GSE253747. ChIP-seq data were deposited in GEO using the dataset identifier GSE255157. The source data are provided in this study. Source data are provided with this paper.

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

## Acknowledgements
The authors thank the patients for their participation and staff at the First Affiliated Hospital, Sun Yat-sen University for their assistance throughout the study. This research was financially supported by the National Key R&D Program of China (2021YFA0805100 to J.S.O.), National Natural Science Foundation of China (Grants 92268202 and 81830013 to J.S.O.; 82270485, 81970363 to Z.J.O.; 82230010 to W.K.), International cooperation project from the Ministry of Science and Technology of China (2015DFA31070 to J.S.O.), Non-profit Central Research Institute Fund of Chinese Academy of Medical Sciences (2023-PT320-03 to Y.L.), Guangdong Basic and Applied Basic Research Foundation (2019B1515120092 to J.S.O.), Science and Technology Planning Project of Guangzhou, China (202103000016 to J.S.O.), Sun Yat-sen University Clinical Research 5010 Program (2014002 to J.S.O.), Science and Technology Planning Project of Guangdong Province (2023B1212060018 to J.S.O.), and Program of National Key Clinical Specialties.

## Author contributions
J.S.O., W.K., and Z.J.O. conceived and designed the study. Z.L.L. and Y.J.L. performed most experiments during the later stages of the study. M.M.S. and M.X.F. performed the primary experiments during the early stages of the study. Y.L. participated in all aspects of the study. Z.Q.L., D.S.N., X.M.Z., X.L., Q.H.C., Y.M.P., X.M.Z., Y.R.H., J.S.L., Y.J.L., and M.W. performed the experiments. Z.L.L., Y.L., Y.J.L., M.M.S., M.X.F., Z.Q.L., D.S.N., X.M.Z., and C.X.Z. analyzed the data. Z.L.L. and J.S.O. prepared the manuscript. J.S.O., W.K., and Z.J.O. supervised the study and revised the manuscript.

## Competing interests
The authors declare no competing interests.

## Additional information

[1]Division of Cardiac Surgery, Cardiovascular Diseases Institute, The First Affiliated Hospital, Sun Yat-sen University, Guangzhou, P.R. China. [2]National-Guangdong Joint Engineering Laboratory for Diagnosis and Treatment of Vascular Diseases, Guangzhou, P.R. China. [3]NHC key Laboratory of Assisted Circulation and Vascular Diseases (Sun Yat-sen University), Guangzhou, P.R. China. [4]Key Laboratory of Assisted Circulation and Vascular Diseases, Chinese Academy of Medical Sciences, Guangzhou, P.R. China. [5]Guangdong Provincial Engineering and Technology Center for Diagnosis and Treatment of Vascular Diseases, Guangzhou, P.R. China. [6]Department of Physiology and Pathophysiology, School of Basic Medical Sciences, Peking University, Beijing, P.R. China. [7]Key Laboratory of Molecular Cardiovascular Science, Ministry of Education, Beijing, P.R. China. [8]Department of Biomedical Informatics, School of Basic Medical Sciences, Peking University, Beijing, P.R. China. [9]Department of Cardiovascular Surgery, The Second Xiangya Hospital of Central South University, Changsha, P.R. China. [10]Division of Vascular Surgery, The First Affiliated Hospital, Sun Yat-sen University, Guangzhou, P.R. China. [11]Department of Pharmacology and Cardiovascular Research Center, Rush Medical College, Rush University Medical Center, Chicago, IL, USA. [12]Department of Cardiology, Institute of Cardiovascular Research, the Affiliated Hospital, Southwest Medical University, Luzhou, China. [13]Division of Hypertension and Vascular Diseases, Department of Cardiology, Cardiovascular Diseases Institute, The First Affiliated Hospital, Sun Yat-sen University, Guangzhou, P.R. China. [14]Guangdong Provincial Key Laboratory of Brain Function and Disease, Zhongshan School of Medicine, Sun Yat-sen University, Guangzhou, P.R. China. [15]These authors contributed equally: Ze-Long Liu, Yan Li, Yi-Jun Lin, Mao-Mao Shi, Meng-Xia Fu. ✉e-mail: kongw@bjmu.edu.cn; Zhijunou@163.com; oujs@mail.sysu.edu.cn; oujs2000@163.com

