## [Peer Review File · Nature Communications]

Aging aggravates aortic aneurysm and dissection via miR-1204—MYLK signaling axisREVIEWER COMMENTS

Reviewer #1 (Remarks to the Author):

In the present manuscript, Liu and colleagues describe the role of miR-1204 in the regulation of aneurysm formation in patients. The authors performed miRNA screening on 430 individuals and studied the identified miRNA in cellular models, elucidating its role in regulating VSMC senescence in a P-53 dependent manner. The study is very well executed and includes several in vivo data using an animal model of aneurysm formation. However, there are some limitations regarding the underlying molecular mechanisms and cellular interactions in this context. Obtaining more information in these areas would significantly enhance the value of this study.

Below are specific points for the authors to consider:

1. Figure S2A: It is not clear how the luciferase signal works with the agomiR labeled with a fluorophore (Cy5). Please provide a detailed explanation of the system. Additionally, the authors should address the surprising observation of massive localization in the aorta and minimal localization in highly vascularized tissues such as the heart and lungs.
2. miR-1204 in vivo modulation: Please show miR-1204 level in the aorta in both loss- and gain-of-function by performing RT-qPCR.
3. Basal expression level: Compare miR-1204 expression in the aorta with that in other cardiovascular tissues at the basal level.
4. Choice of animal model: Explain the rationale behind using C57BL/6 animals, considering that the penetrance of aneurysm formation in this background is known to be limited compared to ApoE^{-/-}.
5. Evaluation of aortic structure: Did the authors evaluate the contractility of the aorta in the ecographic analysis? It would be valuable to include measurements of maximal aortic radial wall velocity using M-mode (DOI: 10.1161/01.hyp.26.1.48).
6. Effect of conditioned medium: Test the effect of conditioned medium from gain-of-function miR-1204 overexpressing VSMCs on immune cells such as macrophages.
7. Evaluation of VSMC features: Assess the effect of miR-1204 on other typical VSMC characteristics, including proliferation, migration, and contractility. This analysis should also be performed for the rescue experiments.
8. The authors must show the overexpression of MYLK by RT-qPCR and western blot.
9. Mature miR-1204 levels: Determine the level of mature miR-1204 in AII-treated VSMCs.
10. Response to VSMC stimuli: Evaluate how miR-1204 expression responds to classical VSMC de-differentiation (PDGF-BB) and differentiation (TGB) stimuli.
11. Comparison with other VSMC miRNAs: Compare the basal expression level of miR-1204 with other important VSMC miRNAs, such as miR-145.
12. In the section discussing p53 activity on miR-1204, it is important for the authors to include the measurement of mature miR-1204 levels. This additional information will provide a more comprehensive understanding of the regulatory effects of p53 on miR-1204.

Addressing these points will enhance the clarity and comprehensiveness of the study, providing a stronger foundation for the molecular mechanisms and cellular interactions involved in aneurysm formation.

Reviewer #2 (Remarks to the Author):

The authors identified miR-1204 as an important factor of AAD in young versus elder patients that showed a positive correlation with age. miR-1204 augmented AngII-induced aortic pathology, and inhibition of miR-1204 attenuated BAPN-induced AAD. In cultured vascular smooth muscle cells (VSMCs), the authors found that miR-1204 directly targeted MYLK to promote senescence-related phenotype. The theme is interesting. While a large amount of data were presented, there are multiple concerns that need to be addressed.

1. Figures 4-7 and S5-6 were VSMC culture analyses. There has no direct evidence that miR-1204 and MYLK in VSMCs contributes to AAD through an aging-related mechanism.
2. Overall the animal studies show significant phenotypes as presented in Figures 2 and 3. However, quality of histological images needs to be improved.
3. Statistical analysis should be specified in figure legends. Also, statistical analysis method section

stated that equal variance was tested, but did not provide how the results of this test was used for data analysis.

4. The authors did not provide necessary information about where (the aortic region) VSMCs were obtained for cell culture. Also, some experiments used AngII in cultured cells. It is unclear how AngII stimulates VSMCs to study TAA, AAA, or AAD because reported in vivo mouse studies failed to demonstrate that AT1 receptor in VSMCs contributes to AngII-induced aortic pathologies in either ascending or abdominal aortic regions.

5. Figure 1C: Did the author compare young normal vs elder normal, and young normal vs young patient? Since miR-1204 is positively correlated with aging in elder patients, it is possible that this represents a potential mechanism of aortic pathologies in elder, but not young patients. However, the animal studies used either young adult mice (4 months old for AngII infusion) or mice just after weaning (for BAPN study). How can these results link to the miR-1204-mediated aging mechanism of AD/TAA/AAA?

6. Figure 1E: What were the comparisons between young versus elder patients for TAAD and AAA, respectively?

7. Figure 2F: Did AngII change the diameters of ascending, descending, and abdominal aortic regions in miR-ctl mice? The data were acquired using Vevo 3100 ultrasound, but no detailed methods were provided. For example, how were data for descending aortas acquired? This location may not be visualized appropriately using a parasternal view.

8. Figure 2J: The comparisons were between non diseased versus pathological tissues. Therefore, SA-beta-gal activity change may not be explained as an aging-mediated mechanism.

9. Table 1 comparisons should be two factors, not one factor; one factor is young versus old; one factor is normal versus patients. Also, young versus older patients should be compared for differences in hypertension and diabetes.

10. There are many grammatical errors that need to be carefully edited. Only provide a couple of descriptions here:

(1) The reason using 50 years old because the average age of AD is less than 50 years old in China, which is significantly different western country.

(2) Accumulating evidences suggests that the SASP participates in various diseases, including ...

Reviewer #3 (Remarks to the Author):

In the manuscript titled "Aging aggravates aortic aneurysm and dissection via miR-1204—MYLK signaling axis" the authors present a well-written study on the molecular mechanisms regulating the overexpression of miR-1204 microRNA in the aging process in VSMCs, and its implication in the formation of AADs.

To do this, the authors perform an ordered battery of experiments in which they first find the increase of miR-1204 in patients with AAD. They then show that expression of miR-1204 aggravates the effects in AAD (inducing a SASP phenotype and loss of contractility), and that its inhibition attenuates the effects of disease in VSMCs. Finally, the authors study the modulating (attenuating) effect of the MYLK kinase on the effects induced by miR-1204, possibly regulated by the TGFB signaling pathway.

Several points need to be commented and clarified.

Major comments:

1- Although this work is fundamentally focused on the study of the molecular mechanisms that are involved in the senescence of VSMCs and in the aggravation of AADs, the authors comment on the potential of the expression of miR-1204 as a biomarker for age-related AADs. I would like to know how other previously described markers, such as miR-140 5p or miR-140 3p, behave in the VSMC models generated for this study, and how the diagnostic power of miR-1204 compares with other microRNAs. Likewise, the expression analysis of miR-1204 should be studied in an independent validation cohort.

2- In the study, a correlation is performed between the expression levels of miR-1204 and age, which results in a low, although significant, R2 value. However, this study is only done in patients. Considering that miR-1204 expression is also markedly higher in the elder group of control participants, is there also a significant correlation between miR-1204 expression and age in the

control group? If so, could the expression of miR-1204 be considered more age- or disease-specific?

3- In addition to the miR-1204 microRNA, another microRNA (miR-124 3p) was differentially expressed in the comparative study between the elderly patient and control groups. Although the authors consider the expression of this microRNA to be aberrant, which apparently coincides with another cited study, it is not clear to this reviewer what are the reasons for discarding this microRNA from the study.

Furthermore, the authors of the cited work determined the expression of miR-124 as a critical regulator in human aortic VSMC differentiation by targeting the 3-UTR of Sp1, and they present a panel of markers identical (SMA, SM22a, Calponin and MYH11) to the one used in the present study to determine the phenotype of the VSMCs.

4- In this study, a SIL-based proteomics study has been carried out between control VSMCs and cells expressing miR-1204.

Several questions about this study must be answered/commented:

a- How many proteins were identified and quantified in the study?

b- the chosen cut of fold change (1.3) seems too low.

c- What are the quantification data for important proteins such as SIRT1, MRTFB or KLF5 whose expression is clearly affected in AADs?

d- What are the quantitative data of the kinase MYLK? Can its decrease be detected in cells expressing miR-1204 in comparison with the control cells?

5- In the TargetScan study for miR-1204 several candidate genes were identified, one of which is MYLK, however, two other genes, RHOJ and DHCR7, appeared as targets for miR-1204. Although these two genes are not specific to VSMCs, how could the authors explain their importance in the molecular mechanism of miR-1204?, and how could they fit into the hypothetical working model presented in Fig7F?

It is necessary to make a detailed comment in the discussion.

6- The authors comment in the manuscript "Although miR-1204 agomir did not affect AAD formation and survival, it remarkably aggravated AngII-induced AAD incidence and mortality (Figure 2B-D)". I agree, looking at Figure 2B, that there is a notable increase in incidence, however, it does not seem that this increase is correlated with a notable increase in mortality, which would be very slight. How can the authors better explain this result?

minor comments:

1- In an epigraph of the work it is written: "MYLK ameliorates miR-1204 induced phenotype switch possibly via transforming growth factor- β (TGF- β) signaling pathway in VSMCs". In my opinion, the word "possibly" should not appear in a results heading, since that would indicate that there are other signaling pathways that could explain the same results and that would be another aspect that could be studied in another results heading. In this case, other options could be cited/developed in a discussion paragraph.

2- The authors do not show, or at least this reviewer has not found, the results that demonstrate the inhibition of miR-1204 expression in aorta in the LNA anti-miRNA experiment.

3- In Figure 1E, is there a significant change between the TAAD and AAA? How could this difference be explained?

4- there is a typo on page 7, line 102. Change "studied" to "studies"

Reviewer #4 (Remarks to the Author):

In this article, Liu et al. examined the role of the miR-1204-MYLK axis in cellular senescence of

VSMCs and the development of AAD. This study suggests that miR-1204 is increased in senescent VSMCs and especially in aortas associated with AAD in the elderly, upregulating inflammatory factors and promoting phenotypic shifts in VSMCs via suppression of MYLK, thereby contributing to the progression of cellular senescence. It is already known that miR-1204 is regulated by p53 expression and that MYLK is involved in the regulation of inflammation.

What is most difficult for the reviewer to understand is that cellular senescence can be inhibited by altering the expression of miR-1204 and MYLK, which have been suggested to be downstream of p53. Normally, it is difficult to alter the state of cellular senescence by changing p53 target factors alone. Since most of the evaluation of cellular senescence in this study is based on SA-beta-gal activity alone, data on multiple markers of cellular senescence such as p53, p16, p21, and g-H2AX should be shown, as well as the state of cell division if the cells are cultured.

Although it has been reported that AngII promotes cellular senescence, it is unlikely that BAPN, an inhibitor of LOX activity, induces cellular senescence. Under such conditions, the reviewer wonders how anti-miR-1204 could cancel AAD formation in BAPN-treated mice. miR-1204 may have some effect on LOX activity or elastin/collagen cross-linking?

The reviewer also has the following concerns:

It was unclear from which organs and tissues miR-1204 is mainly supplied to the circulatory system and aorta, and how miR-1204 reaches the aortic tunica media. miR-1204 is produced in the aorta and affects the local area? Or is it produced in other organs and transported to the aorta via the circulation?

miR-1204 was increased in the aorta of young patients, but not in the plasma. What is the reason for this discrepancy?

Likewise, why was miR-1204 specifically expressed or accumulated in the aorta? It appears that miR-1204 injected through the tail vein accumulated only in the aorta and not in other organs (Figure S2A). Is there a specific receptor or transport system that delivers miR-1204 specifically to the aortic media?

The authors argued that miR-1204 promotes cellular senescence via inhibition of MYLK. However, treatment with miR-1204 without AngII did not induce cellular senescence in the aorta in Figure 2. I wonder why?

Minor comments:

There are discrepancies between the microscopic images and their quantification data, especially in Figures 1G, 2J, 3H, and 5D.

There are inconsistencies in the levels of miR-1204 in the plasma in Figure 1C and Figure S2C.

Figure 2C was not statistically significant, but the authors state: "significantly worsened Ang-II-induced AAD mortality. Is this statement correct?

In Figure S3A, change the order of the miRs as in miR-1204"-1", "-2", "-3".

Grammatical and spelling errors should be corrected.

Reviewer #1

Thank you for acknowledging the thorough execution of our study.

Comment 1. Figure S2A: It is not clear how the luciferase signal works with the agomiR labeled with a fluorophore (Cy5). Please provide a detailed explanation of the system. Additionally, the authors should address the surprising observation of massive localization in the aorta and minimal localization in highly vascularized tissues such as the heart and lungs.

Response : Thank you for your important comment. We recognize the error that caused this confusion. This should be “fluorescent signal” instead of “luciferase signal.” The Cy5 fluorophore was labeled at the 3’ end of the agomir molecule. The Cy5-labeled miR-1204 agomir was delivered via tail venous injection, and fluorescence imaging was performed after 6 h using an IVIS imaging system (Caliper Life Sciences, MA, USA). The excitation and emission wavelengths were 649 and 680 nm, respectively. We initially examined the accumulation of miR-1204 agomir in the aorta to demonstrate its ability to reach this specific tissue. In the revised manuscript, we have supplemented the study by investigating the distribution of the miR-1204 agomir in other cardiovascular tissues following injection via the tail vein. **As shown in Figure S2E**, the miR-1204 agomir accumulated in the heart, lungs, kidneys, liver, and aorta, indicating nonspecific distribution rather than specific aggregation in the aorta.

Comment 2. miR-1204 in vivo modulation: Please show miR-1204 level in the aorta in both loss- and gain-of-function by performing RT-qPCR.

Response: Thank you for the suggestion. Successful inhibition and overexpression of

miR-1204 *in vivo* were confirmed via RT-qPCR of miR-1204 expression in anti-1204 and miR-1204 agomir-injected mice aortas compared with control mice treated with scr-miRNA (**Please see Figure S2G-H**).

Comment 3. Basal expression level: Compare miR-1204 expression in the aorta with that in other cardiovascular tissues at the basal level.

Response: Thank you for this valuable suggestion. The baseline expression level of miR-1204 was assessed using RT-qPCR. As depicted in **Figure S2A**, miR-1204 exhibited similar expression in various tissues, including the heart, lung, kidney, liver, and aorta in C57BL/6 mice. However, after AngII infusion, miR-1204 is elevated in the aorta compared to the control group (**Please see Figure S2D**).

Comment 4. Choice of animal model: Explain the rationale behind using C57BL/6 animals, considering that the penetrance of aneurysm formation in this background is known to be limited compared to ApoE^{-/-}.

Response: We acknowledge this important point. The choice of animal model was based on the requirements of the experimental design. The incidence of aneurysms has been reported to be quite high in hyperlipidemic mice, sometimes reaching 80-90%^{1,2}. In this case, the potentially detrimental role of miR-1204 in the AngII model may not be fully manifested. The incidence of aneurysms was lower in C57BL/6 mice than in hyperlipidemic mice, which was suitable for our experimental design. In our study, AngII induced aneurysms in three (16%) of 19 C57BL/6 control mice. The incidence was significantly higher in the miR-1204 agomir + AngII mice, in which 10 (53%) of 19 mice developed aneurysms (**Figure 2B**). These findings indicate that miR-1204 is a crucial mediator of Ang II-induced AAD formation.

Therefore, the use of C57BL/6 mice highlighted the importance of miR-1204 in aggravating aortic aneurysms and dissection.

Comment 5. Evaluation of aortic structure: Did the authors evaluate the contractility of the aorta in the ecographic analysis? It would be valuable to include measurements of maximal aortic radial wall velocity using M-mode (DOI: 10.1161/01.hyp.26.1.48).

Response: We are appreciative of the reviewer's suggestion. Indeed, evaluating the contractility of the aorta *in vivo* would be more useful. In accordance with the reviewer's suggestion, additional animal experiments have been conducted in the revised version. **As depicted in following Figure #1**, co-administration of the miR-1204 agomir with AngII treatment significantly reduced M-mode velocity in the abdominal aorta of mice. Additionally, MYLK overexpression demonstrated a pronounced rescue effect. Furthermore, the contraction of aortic rings isolated from AngII-treated mice in response to the vasoconstrictor phenylephrine was measured *ex vivo*. Aortas overexpressing miR-1204 exhibited reduced contractility upon phenylephrine stimulation. In contrast, MYLK overexpression significantly alleviated the miR-1204-induced contractile strength reduction (**Please see Figure 6N**).

Figure #1. Representative M-mode images and quantification of M-mode velocity in
3

the abdominal aortas of AngII-treated mice. n=17–20. The data were presented as the mean±SD and analyzed using one-way ANOVA followed by Tukey's multiple comparisons. *, $P<0.05$; ****, $P<0.0001$.

Comment 6. Effect of conditioned medium: Test the effect of conditioned medium from gain-of-function miR-1204 overexpressing VSMCs on immune cells such as macrophages.

Response : Taking into account the reviewer's suggestion, we have established a culture system featuring PMA-differentiated THP-1 macrophages on the insert membrane, with a conditioned medium added to the well (**Please see Figure S7A**). The conditioned medium from miR-1204 overexpressing VSMCs significantly induced macrophage migration in transwell chambers. However, MYLK overexpression markedly circumvented this effect (**Please see Figure S7B**). In addition, we validated the effect of conditioned medium treatment on the macrophage phenotype. M1-type macrophages promote inflammatory processes. Thus, a representative marker of the M1 phenotype, CD80, was detected using flow cytometry. Compared to the control group cells, conditioned medium from miR-1204 overexpressing VSMCs significantly induced macrophages to polarize towards the M1 phenotype. Similarly, MYLK overexpression attenuated this effect (**Please see Figure S7C**). Taken together, the overexpression of miR-1204 induces the secretion of cytokines by VSMCs, fostering macrophage migration and M1 polarization. Notably, this process was regulated by MYLK.

Comment 7. Evaluation of VSMC features: Assess the effect of miR-1204 on other typical VSMC characteristics, including proliferation, migration, and contractility. This analysis should also be performed for the rescue experiments.

Response: We appreciate your suggestion. To address this, we conducted experiments to elucidate the influence of miR-1204 on VSMC characteristics and presented the findings as follows: Consistent with the impact of miR-1204 on VSMC senescence, overexpression of miR-1204 in VSMCs led to a notable suppression of cell proliferation, as determined using the EdU proliferation assay (**Please see Figure S7D**) and MYLK overexpression blunted the effects of miR-1204 on VSMC proliferation. Furthermore, VSMCs miR-1204 overexpressing exhibited reduced contractility as evidenced by a collagen gel contraction assay, which abolished the effects of miR-1204 on VSMC contractility (**Please see Figure S7E**). Moreover, VSMCs that overexpressed miR-1204 exhibited reduced migration as indicated by scratch-wound assay. Similarly, MYLK overexpression attenuated the inhibitory effect of miR-1204 on VSMC migration (**Please see Figure S7F**). Given that MYLK is a crucial kinase for myosin regulatory light-chain phosphorylation and contractile responses, the effects of miR-1204 and MYLK overexpression on VSMC migration and contractility were expected. Taken together, these findings indicate that miR-1204 inhibits VSMC proliferation, migration, and contractility by regulating MYLK.

Comment 8. The authors must show the overexpression of MYLK by RT-qPCR and western blot.

Response: Thank you for this suggestion. As shown in **Figure S5A-B**, VSMCs were infected with adenovirus particles at various multiplicities of infection (MOI), and the mRNA and protein levels of MYLK were assessed. MYLK expression levels have already exhibited a significant increase at an MOI of 10.

Comment 9. Mature miR-1204 levels: Determine the level of mature miR-1204 in

AII-treated VSMCs.

Response: We have supplemented this part according to the reviewer's suggestion. Similar to previous results of the detection of pri-mir-1204 expression, AngII treatment significantly enhanced mature miR-1204 expression compared to that in untreated control cells (**Please see Figure S9D**).

Comment 10. Response to VSMC stimuli: Evaluate how miR-1204 expression responds to classical VSMC de-differentiation (PDGF-BB) and differentiation (TGB) stimuli.

Response: Thank you for the suggestion. Through RT-qPCR, we found that mature miR-1204 expression was significantly downregulated by TGF- β (**Please see Figure S9B**) but upregulated by PDGF-BB (**Please see Figure S9C**), suggesting that miR-1204 was transcriptionally regulated during VSMC phenotypic modulation.

Comment 11. Comparison with other VSMC miRNAs: Compare the basal expression level of miR-1204 with other important VSMC miRNAs, such as miR-145.

Response: Thank you for the suggestion. Based on the RT-qPCR results, our data indicates that the endogenous expression level of miR-1204 was lower than that of other benchmarked microRNAs in VSMCs, such as the miR143/145 cluster (**Please see Figure S9A**). However, under the influence of vascular aging and disease states, the upregulation of miR-1204 was significant. This change in the cellular level of miR-1204 may affect VSMC properties.

Comment 12. In the section discussing p53 activity on miR-1204, it is important for

the authors to include the measurement of mature miR-1204 levels. This additional information will provide a more comprehensive understanding of the regulatory effects of p53 on miR-1204.

Response: Thank you for the comment. As per your suggestion, we have measured the expression levels of mature miR-1204 in p53-activated VSMCs. Consistent with the expression changes of pri-mir-1204, p53 activation induced an elevation in the expression of miR-1204, and silencing p53 eliminated this effect (**Please see Figure S9E**).

Reviewer #2

We thank the reviewer for your feedback and for taking interest in our study.

Comment 1. Figures 4-7 and S5-6 were VSMC culture analyses. There has no direct evidence that miR-1204 and MYLK in VSMCs contributes to AAD through an aging-related mechanism.

Response : We appreciate your valuable suggestion. To investigate the role of miR-1204 and MYLK in AAD development through aging-related mechanisms, we conducted *in vivo* experiments using a mouse model of AngII-induced AAD. 10 nmol dose of miR-1204 agomir or miR-control was delivered by tail vein injection to C57BL/6 mice every 3 d for a total of five times. Then, C57BL/6 mice were tail vein-injected with MYLK adenovirus or blank adenovirus before a 4-week AngII infusion (**Please see Figure 6H**). The miR-1204 agomir + AngII group showed substantially increased TAD occurrence and SA- β -gal positive staining in contrast with control groups. As expected, the MYLK overexpression group showed a significant decline in the incidence of TAD. Notably, SA- β -gal staining was markedly

suppressed (**Please see Figure 6I-K**). Immunofluorescence staining confirmed effective overexpression of MYLK in the aorta. Additionally, the expression of SASP markers, IL-6 and MCP-1, was increased in the aortic media of the miR-1204 + AngII group, and these effects were attenuated by MYLK overexpression (**Please see Figure 6M**). These results provide direct evidence that increased miR-1204 expression and deregulated MYLK expression participate in AAD formation through aging-related mechanisms.

Comment 2. Overall the animal studies show significant phenotypes as presented in Figures 2 and 3. However, quality of histological images needs to be improved.

Response: Thank you for the suggestion. We have replaced the histological images with clearer ones to enhance the quality of the article.

Comment 3. Statistical analysis should be specified in figure legends. Also, statistical analysis method section stated that equal variance was tested, but did not provide how the results of this test was used for data analysis.

Response : We appreciate your suggestion. Normality and equal variance were assessed for the quantitative results to determine the application of a parametric or nonparametric test. Data that passed the aforementioned tests were analyzed using a parametric test. Data that did not pass the tests for normality and equal variance were analyzed using a nonparametric test. Those that passed the normality tests but not the equal variance tests were analyzed using a parametric test with Welch's correction. Specific statistical methods are described in the figure legends. We have corrected the errors in our previous analysis.

Comment 4. The authors did not provide necessary information about where (the aortic region) VSMCs were obtained for cell culture. Also, some experiments used AngII in cultured cells. It is unclear how AngII stimulates VSMCs to study TAA, AAA, or AAD because reported *in vivo* mouse studies failed to demonstrate that AT1 receptor in VSMCs contributes to AngII-induced aortic pathologies in either ascending or abdominal aortic regions.

Response : Based on the reviewer's suggestion, we consulted the manufacturer regarding the issue of cell origin. Primary human aortic VSMCs were obtained from ScienCell (Cat. No. 6110, Carlsbad, California, USA). According to the manufacturer, human aortic VSMCs are sourced from an unspecified region of the aorta and may contain cells from all over the aorta. To further understand the impact of VSMCs from different regions of the aorta, we isolated primary VSMCs from the mouse ascending and abdominal aortas. SM22 immunofluorescence staining confirmed the purity of the cell isolation (**Please see following Figure #2A**). The basal expression level of miR-1204 was not different in cultured VSMCs obtained from different aortic regions (**Please see following Figure #2B**). miR-1204 overexpression decreased MYLK expression in VSMCs. Consistently, SASP components, including IL-6 and MCP-1, were upregulated at the transcriptional level after miR-1204 overexpression (**Please see following Figure #2C-D**). These findings suggest a similar role for miR-1204 in VSMCs from different sources.

Indeed, as the reviewer suggested, *in vivo* experiments showed that the deletion of AT1R in VSMCs does not affect AngII-induced aortic lesions. However, this does not imply that AngII cannot stimulate the VSMCs. Many studies employ Ang II-induced mouse models to investigate the impact of VSMC-specific gene editing on AAD development^{3,4}. In these models, despite no changes in AT1R levels, VSMC-specific gene editing, such as XBP1 and ST3GAL5 knockouts, has been shown to influence

AngII-induced AAD. This suggests that AngII causes AAD by stimulating VSMCs directly.

Furthermore, numerous studies used AngII to stimulate VSMCs *in vitro* to induce cell senescence and transcriptional activation of inflammatory factors and miRNAs. This approach allowed for the exploration of molecular mechanisms related to the formation of AAD⁵. Consistent with these studies, our study demonstrated that AngII successfully induced VSMC senescence and the release of inflammatory factors, such as IL-6 and MCP-1. This phenomenon was amplified by the overexpression of miR-1204, whereas the overexpression of MYLK rescued this effect (**Please see Figure 6**).

Figure #2. A) Identification of vascular smooth muscle cells (VSMCs) obtained from different aorta regions. B) Quantification of miR-1204 levels in VSMCs obtained from different aorta regions. n=3. C) Quantification of mRNA levels of MYLK, interleukin 6 (IL-6), and monocyte chemotactic protein 1 (MCP-1) in VSMCs from ascending aorta transfected with miR-1204 mimics or miR-control (miR-ctl). n=3. D) Quantification of mRNA levels of MYLK, IL-6, and MCP-1 in VSMCs from abdominal aorta transfected with miR-1204 mimics or miR-ctl. n=3. The data were presented as the mean±SD and analyzed using unpaired two-tailed Student's t test for B-D. *, $P<0.05$; **, $P<0.01$; ***, $P<0.001$; ns, not significant.

Comment 5. Figure 1C: Did the author compare young normal vs elder normal, and young normal vs young patient? Since miR-1204 is positively correlated with aging in elder patients, it is possible that this represents a potential mechanism of aortic pathologies in elder, but not young patients. However, the animal studies used either young adult mice (4 months old for AngII infusion) or mice just after weaning (for BAPN study). How can these results link to the miR-1204-mediated aging mechanism of AD/TAA/AAA?

Response: We understand the reviewer's concern. A comparison between young and elderly normal individuals revealed no significant differences. Similarly, no differences were observed between the young normal and young patients (**Please see Figure 1C**).

The choice of animal model was based on the requirements of the experimental design. The elevated incidence rate observed in the AngII-induced model in elderly or genetically modified mice may overshadow the role of miR-1204 in this situation⁶. Thus, young adult mice were chosen for AngII infusion. Importantly, the administration of miR-1204 agomir alone significantly increased SA- β -gal expression in the vascular walls. Simultaneously, miR-1204 agomir further exacerbated SA- β -gal staining in AngII-infused mice, indicating a strong effect of miR-1204 in inducing cell senescence (**Please see Figure 2J**).

The animal model induced by BAPN was based on the results of previous studies⁷. Three-week-old mice have been reported to be suitable for BAPN-induced TAD formation. As the mice got older, the incidence of BAPN-induced TAD decreased and the restrictive effect of BAPN on mouse growth became more apparent. As previously reported, SA- β -gal positive staining was observed as early as day 14 in BAPN-treated mice⁸. In line with the previous study, SA- β -gal staining was detected in the aortic

media of BAPN-treated mice and miR-1204 deficiency reduced vascular cell senescence in our animal model (**Please see Figure 3H**).

Comment 6. Figure 1E: What were the comparisons between young versus elder patients for TAAD and AAA, respectively?

Response: In response to the reviewer's request, we conducted a comparative analysis of miR-1204 expression between young and elderly patients for both TAAD and AAA (**Please see following Figure #3**). In patients with TAAD or AAA, a significant increase in miR-1204 levels was observed in the elderly group.

Figure #3. The difference in miR-1204 expression in the plasma between young patients and elder patients in thoracic aortic aneurysm and dissection (TAAD) group and abdominal aortic aneurysm (AAA) group was determined by RT-qPCR. n=108 (Young TAAD), 101 (elder TAAD), 8 (Young AAA) and 55 (elder AAA). The data were presented as the mean±SD. Data for the TAAD group were analyzed using Mann-Whitney *U* test. Data for the AAA group were analyzed using unpaired two-tailed Student's *t*-test with Welch's correction. **, $P<0.01$; ***, $P<0.001$.

Comment 7. Figure 2F: Did AngII change the diameters of ascending, descending, and abdominal aortic regions in miR-ctl mice? The data were acquired using Vevo 3100 ultrasound, but no detailed methods were provided. For example, how were data for descending aortas acquired? This location may not be visualized appropriately using a parasternal view.

Response : We appreciate your valuable suggestion. Based on our data, AngII treatment had minimal impact on the diameters of the ascending, descending, and abdominal aortic regions in miR-ctl mice, which is consistent with previous studies⁹. However, in the three miR-ctl mice that developed AAA, the diameter of the abdominal aorta was markedly larger than that in the other group.

Noninvasive ultrasound has been widely used to monitor aortic dilation in mice. Many studies have employed ultrasonography to evaluate aortic diameters at various locations, including the ascending aorta, aortic arch, descending aorta, and abdominal aorta^{10,11}. Ultrasonography was performed using a Vevo 2100 ultrasound system equipped with an MS400 transducer (30 MHz). The mice were anesthetized with isoflurane and placed on a heated platform in the supine position. The right parasternal approach was chosen as the standard ultrasound mode for mice. The transducer was placed on the right edge of the sternum at a 45-degree angle relative to the chest.

As the reviewer suggested, this position may not be optimal for descending aortic imaging. To address this concern, we adjusted the probe slightly along the direction of the descending aorta. As illustrated in **following Figure #4**, imaging of the aortic root was hindered, but clear visualization of the descending aorta was achieved in the long-axis views from the aortic arch to the distal regions. Pulse wave mode and color Doppler were employed to ensure the accuracy of our data capture.

Figure #4. A) Representative long-axis right parasternal ultrasound images of non-aneurysmal descending aortas. SA, subclavian artery. B) Color Doppler ultrasound images of the descending aorta. C) Pulse wave mode images of the descending aorta.

Comment 8. Figure 2J: The comparisons were between non diseased versus pathological tissues. Therefore, SA-beta-gal activity change may not be explained as an aging-mediated mechanism.

Response : Thank you for pointing out this issue. For the characterization of cross-sections, aortic sections were collected serially from the proximal to the distal aorta. Histology was determined in the sections that were taken from the aorta root at intervals of 500 μm . Tissue sections from the same cross-sectional level of the aorta in the different groups were for staining analysis and comparison. We have improved the quality of images for better visualization of the SA- β -gal staining in the aorta media (**Please see Figure 2J**).

Comment 9. Table 1 comparisons should be two factors, not one factor; one factor is

young versus old; one factor is normal versus patients. Also, young versus older patients should be compared for differences in hypertension and diabetes.

Response: Thank you for the suggestions. The comparison in Table 1 reveals the differences between the normal and patients. In the revised version, we have added supplemental Table 1 to demonstrate the differences between the young and the elderly participants. As shown in **Supplemental Table 1**, elder patients had a higher incidence of hypertension and diabetes, whereas connective tissue diseases were more prevalent in younger patients.

Comment 10. There are many grammatical errors that need to be carefully edited. Only provide a couple of descriptions here:

(1) The reason using 50 years old because the average age of AD is less than 50 years old in China, which is significantly different western country.

(2) Accumulating evidences suggests that the SASP participates in various diseases, including ...

Response: Thank you. We apologize for these grammatical errors and have corrected them. The revised manuscript was edited by a native English speaker. These sentences have been modified as follows:

a- “ We selected the age of 50 years because the average age of patients with AD in China is less than 50, which is significantly different from that in Western countries (> 60 years)”.

b- “Accumulating evidence suggests...”

Reviewer #3

We thank the reviewer for stating that our study is well-written.

Comment 1 Although this work is fundamentally focused on the study of the molecular mechanisms that are involved in the senescence of VSMCs and the aggravation of AADs, the authors comment on the potential of the expression of miR-1204 as a biomarker for age-related AADs. I would like to know how other previously described markers, such as miR-140 5p or miR-140 3p, behave in the VSMC models generated for this study, and how the diagnostic power of miR-1204 compares with other microRNAs. Likewise, the expression analysis of miR-1204 should be studied in an independent validation cohort.

Response: We appreciate the reviewer's comment. miR-140-5p/-3p is associated with the development and progression of aging-related diseases, including osteoarthritis, osteoporosis, renal fibrosis, ischemic conditions, and Alzheimer's disease. However, there are no reports on their use as circulating biomarkers of aneurysmal diseases. As the reviewer suggested, we have investigated the impact of miR-140-5p/-3p on the phenotype of VSMCs and their ability to induce SASP in our cell model. As shown in **following Figure #5**, the overexpression of miR-140-5p/-3p did not affect the expression of VSMC contractile markers in the presence or absence of AngII stimulation. Similarly, overexpression of miR-140-5p/-3p in VSMCs had minimal impact on SASP expression at the transcriptional level.

Based on the current experimental design, we measured the different expressions of miR-1204 in the normal and diseased states. Further research on its diagnostic value as well as comparisons with other reported miRNAs may require a larger sample size, strict case and control group definitions, sample acquisition protocols, and

standardized miRNA expression detection techniques. We agree that an independent validation cohort would be more convincing. However, obtaining a sufficient number of specimens from patients with AAD within 6 months is challenging, especially when stratifying by age is required. This was added as a limitation of the study. We suggested that further studies should employ a larger sample size.

Figure #5. A-B) Immunoblot analysis of contractile markers, α -SMA and SM22, in VSMCs transfected with a miR-140-3p/-5p mimic or miR-ctl combined with AngII treatment. C) Quantification of mRNA levels of SASP components in VSMCs transfected with miR-140-3p/-5p mimics or miR-ctl. n=6. The data were presented as the mean \pm SD and analyzed using one-way ANOVA.

Comment 2 In the study, a correlation is performed between the expression levels of miR-1204 and age, which results in a low, although significant, R² value. However, this study is only done in patients. Considering that miR-1204 expression is also markedly higher in the elder group of control participants, is there also a significant correlation between miR-1204 expression and age in the control group? If so, could the expression of miR-1204 be considered more age- or disease-specific?

Response : We appreciate the reviewer's consideration of this issue. We also

conducted a correlation analysis between the expression levels of miR-1204 and age in the control group, but no significant correlation was observed ($R^2=0.01378$, $P=0.1419$). This result further validated the specific elevation of miR-1204 expression in older patients.

Comment 3 In addition to the miR-1204 microRNA, another microRNA (miR-124 3p) was differentially expressed in the comparative study between the elderly patient and control groups. Although the authors consider the expression of this microRNA to be aberrant, which apparently coincides with another cited study, it is not clear to this reviewer what are the reasons for discarding this microRNA from the study.

Furthermore, the authors of the cited work determined the expression of miR-124 as a critical regulator in human aortic VSMC differentiation by targeting the 3-UTR of Sp1, and they present a panel of markers identical (SMA, SM22a, Calponin and MYH11) to the one used in the present study to determine the phenotype of the VSMCs.

Response : Thank you for your comment. We did not initially exclude miR-124. Plasma miRNA verification based on our cohort was conducted to validate the expression of miR-124 in different age groups. As shown in **following Figure #6**, although miR-124 was significantly downregulated in patients compared to normal subjects, the pattern of miR-124 changes with age was not apparent. Additionally, the involvement of miR-124 in regulating the VSMC phenotypic switch has been documented. Given that our objective was to identify new age-related miRNAs and elucidate their molecular mechanisms in AAD development, we ultimately decided to exclude the investigation of miR-124.

Figure #6. miR-124 expression in plasma was determined by qRT-PCR in the indicated groups. The data were presented as the mean \pm SD and analyzed using Kruskal-Wallis test with Dunn's multiple comparisons test. *, $P < 0.05$.

Comment 4 In this study, a SIL-based proteomics study has been carried out between control VSMCs and cells expressing miR-1204.

Several questions about this study must be answered/commented:

- a- How many proteins were identified and quantified in the study?
- b- the chosen cut of fold change (1.3) seems too low.
- c- What are the quantification data for important proteins such as SIRT1, MRTFB or KLF5 whose expression is clearly affected in AADs?
- d- What are the quantitative data of the kinase MYLK? Can its decrease be detected in cells expressing miR-1204 in comparison with the control cells?

Response: We appreciate that you pointed this out. Proteomic datasets were deposited in a public database, and the specific responses regarding the mass spectrometry results are as follows:

- a- There were 6,519 proteins identified and 5,581 were quantified in the study.

b- Regarding the criteria for selecting differentially expressed proteins, we chose a 1.3-fold change as the moderate inclusion criterion, which was more suitable for our functional analysis. In addition, our study employed TMT-labeled quantitative proteomics (instead of SILAC), and the differential multiples based on this technique have been commonly reported to be approximately 1.2-fold in many articles^{12,13}.

c- Quantification data for SIRT1 are shown in following **Figure #7 and Table #1**. MRTFB and KLF5 were not detected by mass spectrometry. Failure to identify proteins is common, often due to factors such as low abundance, specific physicochemical properties of proteins, or overlooking protein modifications. As reported earlier, MRTFA expression increases in AAD tissues, and its involvement has been observed in VSMC senescence and AAA formation¹⁴. Therefore, we also measured MRTFA levels using mass spectrometry. The protein levels of SIRT1 and MRTFA showed no significant changes in cells expressing miR-1204 compared to those in control cells, indicating that the impact of miR-1204 on VSMCs was not achieved through the regulation of previously reported genes such as SIRT1 or MRTFA.

d- The quantification data for MYLK are shown in following **Figure #7 and Table #1**. A decrease in MYLK expression was detected in cells expressing miR-1204 compared to control cells.

Figure #7. Heat map of MYLK, SIRT1, and MRTFA expression in overexpressing miR-1204 and control VSMCs.

Table #1. Quantification data for MYLK, SIRT1, and MRTFA in mass spectrometry analysis

Gene name	miR1204 -1	miR1204 -2	miR1204 -3	NC1	NC2	NC3	P
MYLK	0.935	0.854	0.978	1.176	1.045	1.029	0.028
SIRT1	0.976	1.026	0.976	1.028	0.93	1.006	0.55
MRTFA	0.961	1.025	1.029	0.99	0.976	1.026	0.605

Comment 5 In the TargetScan study for miR-1204 several candidate genes were identified, one of which is MYLK, however, two other genes, RHOJ and DHCR7, appeared as targets for miR-1204. Although these two genes are not specific to VSMCs, how could the authors explain their importance in the molecular mechanism of miR-1024? and how could they fit into the hypothetical working model presented in Fig7F?

It is necessary to make a detailed comment in the discussion.

Response: We agree with this point and have expanded on this in the Discussion Section. As suggested, we have added the following content:

“ In addition to MYLK, miR-1204 also markedly suppresses the transcriptional expression of RHOJ and DHCR7. RHOJ, a member of the RHO GTPase family, exhibits highly endothelial cell-restricted expression patterns in several different tissues¹⁵. Relevant studies have demonstrated that RHOJ localizes to focal adhesions and regulates endothelial cell migration and tube formation, while also modulating actomyosin contractility and the number of focal adhesions¹⁶. To date, numerous

studies have documented the crucial causal involvement of endothelial barrier function in the progression of AAD, encompassing disruptions in tight junctions and adherens junctions, as well as extracellular attachments like focal adhesions¹⁷. Despite the low expression levels of RHOJ in aortic VSMCs, future studies should focus on the regulation of RHOJ by miR-1204 in endothelial cells, thereby further exploring the impact of miR-1204 on endothelial barrier function and its role in AAD. DHCR7 catalyzes the conversion of 7-dehydrocholesterol to cholesterol, which is the final step in cholesterol synthesis and significantly affects cellular cholesterol homeostasis¹⁸. The loss of enzymatic activity results in the accumulation of the substrate 7-dehydrocholesterol, which increases the production of vitamin D. The inhibition of dehydrocholesterol reductase has been shown to decrease cholesterol biosynthesis in VSMCs, leading to impaired adhesion, migration, and proliferation, thus affecting their functional steady state¹⁹. Considering that the impact of cholesterol biosynthesis on the characteristics of VSMCs is largely unknown, the influence of miR-1204 on VSMCs in this context deserves further attention.”

Comment 6 The authors comment in the manuscript "Although miR-1204 agomir did not affect AAD formation and survival, it remarkably aggravated AngII-induced AAD incidence and mortality (Figure 2B-D)". I agree, looking at Figure 2B, that there is a notable increase in incidence, however, it does not seem that this increase is correlated with a notable increase in mortality, which would be very slight. How can the authors better explain this result?

Response: Thank you for pointing this out. Considering that AngII treatment is known to have limited efficacy in inducing aneurysm formation and causing aortic rupture-related death in C57BL/6 mice¹, the administration of miR-1204 agomir facilitated AngII-induced AAD incidence and led to premature death. This suggests a

crucial role for miR-1204 in the promotion of AngII-induced AAD. However, there is indeed no significant difference between the two groups based on survival analysis. We have rewritten this part as “Although miR-1204 agomir did not affect AAD formation or survival, it remarkably aggravated AngII-induced AAD incidence, leading to mouse death, a phenomenon not observed in the group receiving AngII alone (**Figure 2B-D**).” In accordance with the reviewer’s suggestion, we have made our description of the results more precise.

Comment 7 In an epigraph of the work it is written: "MYLK ameliorates miR-1204 induced phenotype switch possibly via transforming growth factor- β (TGF- β) signaling pathway in VSMCs". In my opinion, the word "possibly" should not appear in a results heading, since that would indicate that there are other signaling pathways that could explain the same results and that would be another aspect that could be studied in another results heading. In this case, other options could be cited/developed in a discussion paragraph.

Response: As per the reviewer's suggestion, we have removed "possibly" in the result heading.

Comment 8 The authors do not show, or at least this reviewer has not found, the results that demonstrate the inhibition of miR-1204 expression in the aorta in the LNA anti-miRNA experiment.

Response: Thank you for pointing out this issue. We performed a new experiment to show the effective inhibition of miR-1204 *in vivo*. This was verified through RT-qPCR measurements of miR-1204 expression in the aorta of mice injected with LNA anti-1204, compared with control mice treated with scr-miRNA (**Please see the**

new Figure S2H).

Comment 9 In Figure 1E, is there a significant change between the TAAD and AAA? How could this difference be explained?

Response: Thank you for the comment. Considering the distinct characteristics of the different types of AAD, we attempted to detect differences in the plasma levels of miR-1204 between patients with TAAD and those with AAA. As shown in **Figure 1E**, there was no significant difference in the expression of miR-1204 between TAAD and AAA, indicating that age-related pathological mechanisms mediated by miR-1204 are common and may exist in all types of aortic dissections.

Comment 10 there is a typo on page 7, line 102. Change "studied" to "studies"

Response: Thank you for your careful review of our manuscript. We have corrected "studied" to "studies" accordingly.

Reviewer #4

Comment 1 In this article, Liu et al. examined the role of the miR-1204-MYLK axis in cellular senescence of VSMCs and the development of AAD. This study suggests that miR-1204 is increased in senescent VSMCs and especially in aortas associated with AAD in the elderly, upregulating inflammatory factors and promoting phenotypic shifts in VSMCs via suppression of MYLK, thereby contributing to the progression of cellular senescence. It is already known that miR-1204 is regulated by p53 expression and that MYLK is involved in the regulation of inflammation.

What is most difficult for the reviewer to understand is that cellular senescence can be inhibited by altering the expression of miR-1204 and MYLK, which have been suggested to be downstream of p53. Normally, it is difficult to alter the state of cellular senescence by changing p53 target factors alone. Since most of the evaluation of cellular senescence in this study is based on SA-beta-gal activity alone, data on multiple markers of cellular senescence such as p53, p16, p21, and g-H2AX should be shown, as well as the state of cell division if the cells are cultured.

Response: We thank the reviewer for raising this important point.

First, the doses used in our experiments exceeded the physiological levels, which may have produced a significant effect. As shown in the figures (**Please see Figures S2H, S5A-B**), the inhibition of miR-1204 or overexpression of MYLK was quite substantial. Therefore, the significant rescue effects observed are reasonable.

Although our experiments and previous studies have confirmed that miR-1204 is a direct p53 target gene that induces cell senescence, it is important to note that miR-1204 may also regulate the levels of different proteins, introducing complexity into the p53 network. Many studies have illustrated the alteration of cell states by directly regulating the target genes of p53²⁰. For instance, miR-34 is a direct and conserved p53 target gene that presumably mediates the induction of apoptosis, cell cycle arrest, and senescence by p53²¹. Following treatment with DNA-damaging agents, expression of miR-34a and miR-34b/c was induced in a p53-dependent manner. Intriguingly, inhibition of miR-34a through the introduction of miR-34a-specific LNAs resulted in a reduced rate of apoptosis induced by DNA damage²². Similarly, lincRNA-p21, a key regulator of cell proliferation and apoptosis during atherosclerosis is a transcriptional target of p53²³. Inhibition of lincRNA-p21 by recombinant lentivirus vector expressing lincRNA-p21 siRNA increased Ki67

positive cells and decreased apoptosis in a classic murine carotid artery injury model without affecting the expression levels of p53.

In addition to regulating p53 expression, there are multiple layers of molecular mechanisms governing p53 regulation. These include p53 post-translational modifications, temporal expression dynamics, and interactions with cofactors, all of which can influence cell fate²⁴. Therefore, in the revised manuscript, ChIP-Seq assays were performed using p53 antibodies in VSMCs after miR-1204 overexpression. KEGG analysis indicated that many target genes binding to p53 were enriched in pathways related to cell senescence and cell cycle in VSMCs overexpressing miR-1204 (**Please see Figure 7I**). We compared the peak distributions of SASP marker genes including MCP-1, CXCL1, CXCL8, and IGFBP3, in control and miR-1204 overexpression cells and found that miR-1204 overexpression dramatically increased the association of p53 with SASP marker genes (**Please see Figure 7J**). Similarly, induction and activation of p53 involves mainly the uncoupling of p53 from its negative regulators, principally MDM2²⁵. Thus, we examined the phosphorylation status of MDM2 and the binding between MDM2 and p53 in our cellular model. As expected, the phosphorylation of MDM2 was inhibited, and, simultaneously, its binding to p53 was weakened in miR-1204 overexpressing VSMCs. In contrast, MYLK overexpression increased the MDM2/p53 interaction and MDM2 phosphorylation (**Please see Figure 7K-L**). These observations suggest that miR-1204, a transcriptional target of p53, can feed back and regulate the activity of p53, at least in part, by modulating the interaction between p53 and MDM2.

Finally, as suggested, several key senescence marker proteins, including p53, p16, p21, and γ -H2AX, as well as the state of cell division were measured in our cell model. We confirmed that miR-1204 overexpression in VSMCs increased the protein levels of p16, p21, and phosphorylated γ -H2AX, with no change in the expression of p53 protein (**Please see Figure 7E, F**). Conversely, MYLK overexpression inhibited the

expression of these genes. Moreover, miR-1204 substantially inhibited the division of cultured VSMCs (**Please see Figure G, H**). Increased cell division in these cells was further confirmed with MYLK overexpression.

Comment 2 Although it has been reported that AngII promotes cellular senescence, it is unlikely that BAPN, an inhibitor of LOX activity, induces cellular senescence. Under such conditions, the reviewer wonders how anti-miR-1204 could cancel AAD formation in BAPN-treated mice. miR-1204 may have some effect on LOX activity or elastin/collagen cross-linking?

Response: Thank you for your consideration. BAPN could disrupt the crosslinking of the extracellular matrix in the aortic wall, exposing the vascular wall to excessive mechanical stress, and causing TAAD. As previously reported, excessive mechanical stretch could induce VSMC senescence and SA- β -gal positive staining was observed as early as day 14 in BAPN-treated mice⁸. Moreover, the aortas of BAPN-induced TAAD mice exhibited elevated expression of senescence-related proteins, including p53, p19, and p21. These results established a connection between VSMC senescence and BAPN-induced TAAD development. Furthermore, the researcher reported that zoledronic acid treatment reduced the proportion of SA- β -gal positive cells in mouse aorta after BAPN administration, subsequently suppressing BAPN-induced TAAD formation and rupture⁸.

In line with the previous study, SA- β -gal staining was detected in the aortic media of BAPN-treated mice and miR-1204 deficiency reduced vascular cell senescence in our animal model (**Please see Figure 3H**). Our data indicate that VSMC senescence contributes to BAPN-induced TAAD formation and that miR-1204 inhibition provides insights into a potential treatment for TAAD.

Finally, based on the reviewer's suggestion, the LOX activity in cultured VSMCs and

tissue lysates was measured using a high-sensitivity fluorescence assay in this revised manuscript. As illustrated in **following Figure #8**, LOX activity in cultured VSMCs was significantly reduced by BAPN treatment. However, miR-1204 overexpression did not affect LOX activity. Similarly, no difference in LOX activity was detected in the aortas of mice injected with miR-1204 agomir and those injected with scrambled miRNA.

Figure #8. A) Lysyl oxidase (LOX) activity in cultured VSMCs. n=3. The data were presented as the mean±SD and analyzed using one-way ANOVA followed by Tukey's multiple comparisons. B) LOX activity in tissue lysates. n=3. The data were presented as the mean±SD and analyzed using unpaired 2-tailed Student's t test. ****, $P<0.0001$. ns, not significant.

Comment 3 It was unclear from which organs and tissues miR-1204 is mainly supplied to the circulatory system and aorta, and how miR-1204 reaches the aortic tunica media. miR-1204 is produced in the aorta and affects the local area? Or is it produced in other organs and transported to the aorta via the circulation?

Response: We appreciate the reviewer's concern. To address this question, the baseline expression level of miR-1204 in major organs was assessed through RT-qPCR in this revised manuscript. As depicted in **Figure S2A**, miR-1204 exhibited expression in various tissues, including the heart, lung, kidney, liver, and aorta in C57BL/6 mice. Furthermore, we conducted examinations to assess the variations in miR-1204 levels across different organs of mice under saline or AngII infusion. The significant elevation of miR-1204 in the aorta itself under AngII stimulation suggests that local production within the vascular wall is an important factor contributing to the increase in miR-1204 (**Please see Figure S2D**). Nevertheless, in our cultured cell model, we also confirmed that AngII stimulation can enhance the production of pri-mir-1204 and mature miR-1204 levels in VSMCs (**Please see Figures 7A and S9D**). However, based on the current data, we can not exclude the possibility of miR-1204 originating from other organs influencing the aorta. As shown in **the new Figure S2B-D**, miR-1204 is elevated in the white blood cells, heart, and plasma of mice after AngII infusion. Given that we have demonstrated the ability of intravenously injected miR-1204 to adhere to the aorta and exert its effects, we believe that miR-1204 from other tissues or organs can also be released into the bloodstream and reach the vascular wall. In summary, the local elevation of miR-1204 in AAD lesions may result from multiple organ sources, with local production in the vascular wall being a potential primary contributor.

Comment 4 miR-1204 was increased in the aorta of young patients, but not in the plasma. What is the reason for this discrepancy?

Response: Thank you for this question. In many studies, there are inconsistencies between local miRNA and plasma levels. For instance, a study by Kin et al. revealed

that the expression of microRNAs, such as miR-29b, miR-124a, miR-155, and miR-223, which were up-regulated in AAA tissue, were significantly reduced in the plasma of patients with AAA compared to healthy controls and patients with coronary artery disease²⁶. The possible reason is that, apart from plasma, miRNA may also be carried by other components in the blood, such as platelets, peripheral blood mononuclear cells, and extracellular vesicles. Moreover, circulating levels of vascular and inflammation-associated miRNAs could be delivered to dissection lesions, contributing to the reduction of circulating miRNAs in patients with AAD. Based on our data, the changes in the levels of miR-1204 in the younger patient group are not as pronounced as those in the older patient group. This indicates that miR-1204 is an age-related pathogenic miRNA.

Comment 5 Likewise, why was miR-1204 specifically expressed or accumulated in the aorta? It appears that miR-1204 injected through the tail vein accumulated only in the aorta and not in other organs (Figure S2A). Is there a specific receptor or transport system that delivers miR-1204 specifically to the aortic media?

Response: Thank you for the comment. In previous experiments, we examined the accumulation of the miR-1204 agomir in the aorta to demonstrate its ability to reach this specific tissue. In this study, we investigated the distribution of the miR-1204 agomir in other organs following tail venous injection. As shown in the **new Figure S2E**, the miR-1204 agomir accumulated in the heart, lungs, kidneys, liver, and aorta, indicating a non-specific distribution rather than specific aggregation in the aorta. miRNAs may be carried in the blood by both the liquid component (plasma) and formed elements (mainly peripheral blood leukocytes). After AngII infusion, we detected increased levels of miR-1204 in both the plasma and peripheral blood mononuclear cells of mice compared with those in the control group, indicating that

miR-1204 is non-specifically carried in the bloodstream.

Comment 6 The authors argued that miR-1204 promotes cellular senescence via inhibition of MYLK. However, treatment with miR-1204 without AngII did not induce cellular senescence in the aorta in Figure 2. I wonder why?

Response: Thank you for the comment. As illustrated in the representative images and statistical graph of SA- β -gal positive staining area in **Figure 2J**, in saline-infused groups, an enlargement of SA- β -gal positive regions in the aortas of miR-1204 overexpressing mice were detected compared with control mice. We have improved the quality of images to better visualize the SA- β -gal staining in the aorta media.

Comment 7 There are discrepancies between the microscopic images and their quantification data, especially in Figures 1G, 2J, 3H, and 5D.

Response: Thank you for pointing this out. We have enhanced the image quality to ensure better alignment of the statistical outcomes with representative images.

Comment 8 There are inconsistencies in the levels of miR-1204 in the plasma in Figure 1C and Figure S2C.

Response: Thank you for your careful review of our manuscript. This inconsistency was due to the choice of different data presentation methods. For plasma miRNA verification of all 430 subjects, data are presented as the difference values relative to the young healthy groups. For mouse plasma miR-1204 detection in **Figure S2C (Figure S2B in the new revision version)**, the data are presented as fold change relative to the control groups.

Comment 9 Figure 2C was not statistically significant, but the authors state: "significantly worsened Ang-II-induced AAD mortality. Is this statement correct?"

Response: We appreciate the reviewer's feedback. We have reviewed the manuscript and confirmed that the term "significantly" was not used when describing the mortality rate. Nevertheless, we appreciate the reviewer's feedback and have revised this section to provide a more accurate description.

Comment 10 In Figure S3A, change the order of the miRs as in miR-1204"-1", "-2", "-3".

Response: Thank you for bringing this to our attention. We have rearranged the order of the miRs in Figure S3A as requested.

Comment 11 Grammatical and spelling errors should be corrected.

Response: We apologize for these errors. The manuscript has been thoroughly revised and edited by a native English speaker.

Reference

1. Daugherty A, Cassis LA. Mouse models of abdominal aortic aneurysms. *Arterioscler Thromb Vasc Biol.* 2004;24:429-434. doi: 10.1161/01.ATV.0000118013.72016.ea
2. Deng GG. et al. Urokinase-type plasminogen activator plays a critical role in

- angiotensin II-induced abdominal aortic aneurysm. *Circ Res.* 2003;92:510-517. doi: 10.1161/01.Res.0000061571.49375.E1
3. Zhao GZ. et al. Unspliced XBP1 Confers VSMC Homeostasis and Prevents Aortic Aneurysm Formation via FoxO4 Interaction. *Circ Res.* 2017;121:1331-1345. doi: 10.1161/circresaha.117.311450
 4. Zhang F. et al. Ganglioside GM3 Protects Against Abdominal Aortic Aneurysm by Suppressing Ferroptosis in Vascular Smooth Muscle Cells. *Circulation.* 2023. doi: 10.1161/circulationaha.123.066110
 5. Chen HZ. et al. Age-Associated Sirtuin 1 Reduction in Vascular Smooth Muscle Links Vascular Senescence and Inflammation to Abdominal Aortic Aneurysm. *Circ Res.* 2016;119:1076-1088. doi: 10.1161/circresaha.116.308895
 6. Sawada H, Lu HS, Cassis LA, Daugherty A. Twenty Years of Studying AngII (Angiotensin II)-Induced Abdominal Aortic Pathologies in Mice: Continuing Questions and Challenges to Provide Insight Into the Human Disease. *Arterioscler Thromb Vasc Biol.* 2022;42:277-288. doi: 10.1161/atvbaha.121.317058
 7. Ren WH. et al. β -Aminopropionitrile monofumarate induces thoracic aortic dissection in C57BL/6 mice. *Sci Rep.* 2016;6:28149. doi: 10.1038/srep28149
 8. Zhang WM. et al. Sustained activation of ADP/P2ry12 signaling induces SMC senescence contributing to thoracic aortic aneurysm/dissection. *J Mol Cellular Cardiol.* 2016;99:76-86. doi: 10.1016/j.yjmcc.2016.08.008
 9. Fu Y. et al. Cartilage oligomeric matrix protein is an endogenous β -arrestin-2-selective allosteric modulator of AT1 receptor counteracting vascular injury. *Cell Res.* 2021;31:773-790. doi: 10.1038/s41422-020-00464-8
 10. Luo SS, Ji Y. Endothelial HDAC1-ZEB2-NuRD Complex Drives Aortic Aneurysm and Dissection Through Regulation of Protein S-Sulfhydration. *Circulation.* 2023;148:1415-1416. doi: 10.1161/circulationaha.123.066273
 11. Zhang TT. et al. Bestrophin3 Deficiency in Vascular Smooth Muscle Cells

- Activates MEKK2/3-MAPK Signaling to Trigger Spontaneous Aortic Dissection. *Circulation*. 2023;148:589-606. doi: 10.1161/circulationaha.122.063029
12. Bi XJ. et al. Proteomic and metabolomic profiling of urine uncovers immune responses in patients with COVID-19. *Cell Rep*. 2022;38(3):110271. doi: 10.1016/j.celrep.2021.110271
13. Berezcki E. et al. Synaptic markers of cognitive decline in neurodegenerative diseases: a proteomic approach. *Brain*. 2018;141:582-595. doi: 10.1093/brain/awx352
14. Gao P. et al. MKL1 cooperates with p38MAPK to promote vascular senescence, inflammation, and abdominal aortic aneurysm. *Redox Biol*. 2021;41:101903. doi: 10.1016/j.redox.2021.101903
15. Yuan L. et al. RhoJ is an endothelial cell-restricted Rho GTPase that mediates vascular morphogenesis and is regulated by the transcription factor ERG. *Blood*. 2011;118:1145-1153. doi: 10.1182/blood-2010-10-315275
16. Kaur S. et al. RhoJ/TCL Regulates Endothelial Motility and Tube Formation and Modulates Actomyosin Contractility and Focal Adhesion Numbers. *Arterioscler Thromb Vas Biol*. 2011;31:657-664. doi: 10.1161/atvbaha.110.216341
17. Yang XY. et al. Targeting endothelial tight junctions to predict and protect thoracic aortic aneurysm and dissection. *Eur Heart J*. 2023;44:1248-1261. doi: 10.1093/eurheartj/ehac823
18. Prabhu AV, Luu WN, Li DF, Sharpe LJ, Brown AJ. DHCR7: A vital enzyme switch between cholesterol and vitamin D production. *Prog Lipid Res*. 2016;64:138-151. doi: 10.1016/j.plipres.2016.09.003
19. Kohlhaas J. et al. Endothelial cells control vascular smooth muscle cell cholesterol levels by regulating 24-dehydrocholesterol reductase expression. *Exp Cell Res*. 2021;399(2):112446. doi: 10.1016/j.yexcr.2020.112446
20. Hermeking H. p53 enters the MicroRNA world. *Cancer Cell*. 2007;12:414-418. doi: 10.1016/j.ccr.2007.10.028

21. Bommer GT. et al. p53-mediated activation of miRNA34 candidate tumor-suppressor genes. *Curr Biol.* 2007;17:1298-1307. doi: 10.1016/j.cub.2007.06.068
22. Raver-Shapira N. et al. Transcriptional activation of miR-34a contributes to p53-mediated apoptosis. *Mol Cell.* 2007;26:731-743. doi: 10.1016/j.molcel.2007.05.017
23. Wu GZ. et al. LincRNA-p21 Regulates Neointima Formation, Vascular Smooth Muscle Cell Proliferation, Apoptosis, and Atherosclerosis by Enhancing p53 Activity. *Circulation.* 2014;130:1452-U1464. doi: 10.1161/circulationaha.114.011675
24. Hafner A, Bulyk ML, Jambhekar A, Lahav G. The multiple mechanisms that regulate p53 activity and cell fate. *Nat Rev Mol Cell Biol.* 2019;20:199-210. doi: 10.1038/s41580-019-0110-x
25. Meek DW, Hupp TR. The regulation of MDM2 by multisite phosphorylation-Opportunities for molecular-based intervention to target tumours? *Semin Cancer Biol.* 2010;20:19-28. doi: 10.1016/j.semcancer.2009.10.005
26. Kin K. et al. Tissue- and Plasma-Specific MicroRNA Signatures for Atherosclerotic Abdominal Aortic Aneurysm. *J Am Heart Assoc.* 2012;1(5):e000745. doi: 10.1161/jaha.112.000745

REVIEWERS' COMMENTS

Reviewer #1 (Remarks to the Author):

The author's replies satisfy my previous concerns. I do not have anything more to add.

Reviewer #2 (Remarks to the Author):

No further comments

Reviewer #3 (Remarks to the Author):

The authors of the work have responded adequately to all the comments in the first review. They have corrected the errors and added the necessary information in the corresponding headings. Although it is true that the most important limitation of this work was the one mentioned about being able to validate the results obtained in an independent cohort, this reviewer understands that it is not easy to have a cohort with these characteristics.

Reviewer #4 (Remarks to the Author):

The authors have generally responded to my request, so I have no additional comments.